# Targeting ferroptosis protects against experimental (multi)organ dysfunction and death

Samya Van Coillie[1,2], Emily Van San[1,2], Ines Goetschalckx[3], Bartosz Wiernicki [1,2], Banibrata Mukhopadhyay[4], Wulf Tonnus [5], Sze Men Choi[1,2], Ria Roelandt[1,2,6], Catalina Dumitrascu[7], Ludwig Lamberts[3], Geert Dams[3], Wannes Weyts[8], Jelle Huysentruyt[1,2], Behrouz Hassannia[1,2], Irina Ingold[9,10], Suhas Lele[4], Evelyne Meyer[11], Maya Berg[7], Ruth Seurinck [1,6], Yvan Saeys[1,6], An Vermeulen[12], Alexander L. N. van Nuijs[7], Marcus Conrad [9,13], Andreas Linkermann [5,14], Mohan Rajapurkar[4], Peter Vandenabeele [1,2,15], Eric Hoste[16], Koen Augustyns [7] & Tom Vanden Berghe [1,2,3✉]

The most common cause of death in the intensive care unit (ICU) is the development of multiorgan dysfunction syndrome (MODS). Besides life-supporting treatments, no cure exists, and its mechanisms are still poorly understood. Catalytic iron is associated with ICU mortality and is known to cause free radical-mediated cellular toxicity. It is thought to induce excessive lipid peroxidation, the main characteristic of an iron-dependent type of cell death conceptualized as ferroptosis. Here we show that the severity of multiorgan dysfunction and the probability of death are indeed associated with plasma catalytic iron and lipid peroxidation. Transgenic approaches underscore the role of ferroptosis in iron-induced multiorgan dysfunction. Blocking lipid peroxidation with our highly soluble ferrostatin-analogue protects mice from injury and death in experimental non-septic multiorgan dysfunction, but not in sepsis-induced multiorgan dysfunction. The limitations of the experimental mice models to mimic the complexity of clinical MODS warrant further preclinical testing. In conclusion, our data suggest ferroptosis targeting as possible treatment option for a stratifiable subset of MODS patients.

[1] VIB-UGent Center for Inflammation Research, Ghent, Belgium. [2] Department of Biomedical Molecular Biology, Ghent University, Ghent, Belgium. [3] Department of Biomedical Sciences, University of Antwerp, Antwerp, Belgium. [4] Department of Nephrology, Muljibhai Patel Society for Research in Nephro-Urology, Nadiad, India. [5] Department of Internal Medicine 3, University Hospital Carl Gustav Carus, the Technische Universität Dresden, Dresden, Germany. [6] Department of Applied Mathematics, Computer Science and Statistics, Ghent University, Ghent, Belgium. [7] Department of Pharmaceutical Sciences, Toxicological Centre, University of Antwerp, Antwerp, Belgium. [8] VIB-UGent Center for Medical Biotechnology, Ghent, Belgium. [9] Institute of Metabolism and Cell Death, Helmholtz Zentrum München, German Research Center for Environmental Health, Munich, Germany. [10] Department of Medicine III, Klinikum rechts der Isar, Technical University of Munich, Munich, Germany. [11] Department of Pharmacology, Toxicology and Biochemistry, Ghent University, Merelbeke, Belgium. [12] Department of Bioanalysis, Ghent University, Ghent, Belgium. [13] National Research Medical University, Laboratory of Experimental Oncology, Moscow, Russia. [14] Biotechnology Center, Technische Universität Dresden, Dresden, Germany. [15] Methusalem program, Ghent University, Ghent, Belgium. [16] Intensive Care Unit, Ghent University Hospital; Ghent University, Ghent, Belgium. ✉email: Tom.VandenBerghe@uantwerp.be

Seriously ill patients who suffered a life-threatening event, for instance major trauma, surgery, or infection[1], frequently require intensive care unit (ICU) support. Approximately half of all critically ill patients in the ICU develop multiple organ dysfunction syndrome (MODS)[2], which is responsible for 30% of deaths worldwide[3,4]. The extent of organ dysfunction in critically ill patients is correlated to an increase in plasma catalytic iron[5–7] also known as labile iron or non-transferrin bound iron, which is a transitional pool of both extra- and intracellular iron. This unbound iron is highly reactive and results in the production of free radicals causing cellular damage[8]. As such, catalytic iron is known to induce lipid peroxidation[9], the main characteristic of an iron-dependent type of cell death called ferroptosis[10].

Ferroptosis is a recently recognized necrotic type of cell death[10], characterized by iron-dependent oxidation of poly-unsaturated fatty acids (PUFAs) in cell membranes[11,12], resulting in cell rupture[13,14]. An important protectant against ferroptosis is glutathione peroxidase 4 (GPX4)[15], which is a key antioxidant enzyme detoxifying oxidized phospholipids in cell membranes[16]. Hence, GPX4 inactivation results in the uncontrolled accumulation of lipid peroxides, eventually causing the cell to die[15]. In addition to GPX4, other regulatory mechanisms have recently been discovered controlling the lipophilic redox balance such as ferroptosis suppressor protein 1 (FSP1)[17,18] and GTP cyclohydrolase-1 (GCH1)[19]. A typical characteristic of ferroptosis is that it can be blocked by iron-chelating agents, such as defer-oxamine (DFO), or with lipophilic antioxidants such as ferrostatin-1 (Fer1), liproxstatin-1 (Lip1), or vitamin E[10,20,21]. We and others showed that an excess of iron can be sufficient to induce ferroptosis[22,23]. Considering the correlation between ele-vated serum catalytic iron levels and poor outcome in the context of critical illness, we hypothesize that ferroptosis might be a detrimental factor in MODS. This hypothesis is also underscored by case reports showing the use of iron chelation or the natural lipophilic radical trap vitamin E to treat iron overdose induced MODS[24,25].

In this study, making use of a cohort of ICU patients, we find the severity of multiorgan dysfunction and the probability of death among critically ill patients to associate with both catalytic iron levels and excessive lipid peroxidation. In addition, we show a correlation between catalytic iron and lipid peroxidation levels alike and identify a subpopulation of patients with an increased mortality risk based on elevated levels of lipid peroxidation. Next, we demonstrate the crucial role of ferroptosis in a mouse model of iron overload-induced MODS and illustrate the life-saving potency of our candidate lead ferroptosis inhibitor (UAMC-3203) in both these model and a genetically induced model of ferroptosis-driven acute liver injury.

## Results

**Plasma catalytic iron and malondialdehyde associate with MODS and death**. To investigate the association between cata-lytic iron ($Fe_c$), excessive lipid peroxidation, MODS, and death, the levels of $Fe_c$ and malondialdehyde (MDA), a lipid peroxida-tion degradation product, were retrospectively analyzed in plasma of 176 critically ill adult patients enrolled in a prospective cohort study[26]. In this cohort, the median age was 60 (51–70) years. At enrollment, the median sequential organ failure assessment (SOFA) score was 9 (7–11), with 57% of patients suffering from sepsis and 25% of patients having septic shock. The 30-day mortality rate was 23%. To monitor the dynamic fluctuations in these patients, blood was sampled daily for up to 7 days. We found that the maximum value of $Fe_c$ ($Fe_c^{max}$) per patient showed a significant positive correlation with the SOFA score, reflecting the extent of organ dysfunction (Fig. 1a). The $Fe_c^{max}$ values of

patients who succumbed to their illness were significantly higher than those of surviving patients (Fig. 1b), and higher $Fe_c^{max}$ values were found for septic shock patients compared to sepsis patients (Fig. 1c and Supplementary Fig. 1a–g). Similar to $Fe_c^{max}$ values, the maximum value of MDA ($MDA^{max}$) per patient also showed a significant positive correlation with the SOFA score (Fig. 1d) and was significantly higher in the deceased group than in patients who survived (Fig. 1e). In contrast to $Fe_c^{max}$, we found no association of $MDA^{max}$ values with either sepsis or septic shock (Fig. 1f and Supplementary Fig. 1h–n). In keeping with a stronger association of $MDA^{max}$ than $Fe_c^{max}$ with death, only MDA values were consistently higher in the deceased group when analyzed per day (Fig. Supplementary Fig. 2a–n). A positive correlation between $Fe_c$ and MDA within patients is evident from the $Fe_c$ levels being significantly higher on the day a patient reached $MDA^{max}$ compared to the day of the minimum MDA value ($MDA^{min}$) (Fig. 1g). Interestingly, these $MDA^{max}$ values revealed a bimodal distribution for the deceased patients (Fig. 1h). Stratification of all patients based on the local minimum showed that patients with an $MDA^{max} > 14.25\,\mu M$, representing 24.4% of all patients, had a significantly lower survival probability (Fig. 1i). In fact, within this subgroup, 48% deceased within a 30-day follow-up. A more stringent selection, based on the local maximum of the second peak (i.e. $MDA^{max}$ of 16.9 μM) resulted in even higher mortality risk (Supplementary Fig. 2o, p). These findings were confirmed by a Cox proportional hazards regres-sion analysis where a 2-fold increase in either $MDA^{max}$ or $Fe_c$ on the corresponding day a patient reached $MDA^{max}$ resulted in an increase of the daily hazard of death of respectively 90 and 40%, after adjustment for age and SOFA score (Supplementary Fig. 2q). In summary, these data indicate an association between plasma $Fe_c$, excessive lipid peroxidation, the development of MODS, and increased mortality risk. Patients with septic shock also showed higher maximum levels of $Fe_c$ compared to patients with sepsis, which was not observed for their $MDA^{max}$ values.

**Experimental iron overload induces MODS through ferropto-sis**. To further investigate the in vivo effects of excessive $Fe_c$ levels, an experimental iron overload model in C57BL/6 N mice was set up. Based on human case reports of iron intoxication, intraper-itoneal injection of iron(II) sulfate heptahydrate ($FeSO_4$) was presumed to cause MODS[24,25]. We determined 300 mg/kg $FeSO_4$ to be the minimal dose needed to overrule the systemic buffer capacity and induce multiorgan injury (Supplementary Fig. 3a–d), which is similar to what has been reported in human cases of iron intoxication[25,27,28]. A steady increase in iron was observed in several organs as a function of time, which was most prominent in the ileum, while plasma iron levels peaked shortly after injection and subsequently dropped (Fig. 2a and Supple-mentary Fig. 3e). Various plasma injury markers were elevated within 30 min (min) and all increased further as a function of time (Fig. 2b and Supplementary Fig. 3f). Besides measuring plasma lactate dehydrogenase (LDH) as a general biomarker for necrosis, we also monitored aspartate aminotransferase (AST) and alanine aminotransferase (ALT) to reflect liver injury, crea-tinine (Cr) and urea to monitor kidney function, myoglobin (Mb), creatine kinase (CK) and troponin T to assess (cardiac) muscle injury, and ferritin to investigate iron dysbiosis. Except for Cr and Mb, which peaked at 2 h post-iron overload, all other injury biomarkers peaked at 12 h. The exceptionally high levels of CK mainly originated from skeletal muscle tissue, as opposed to heart or smooth muscle tissue (Supplementary Fig. 4a, b). MDA levels were determined to monitor excessive lipid peroxidation in multiple organs. A rapid increase in MDA levels, peaking at 30 min to 1 h after $FeSO_4$ injection, was observed in kidney, liver,

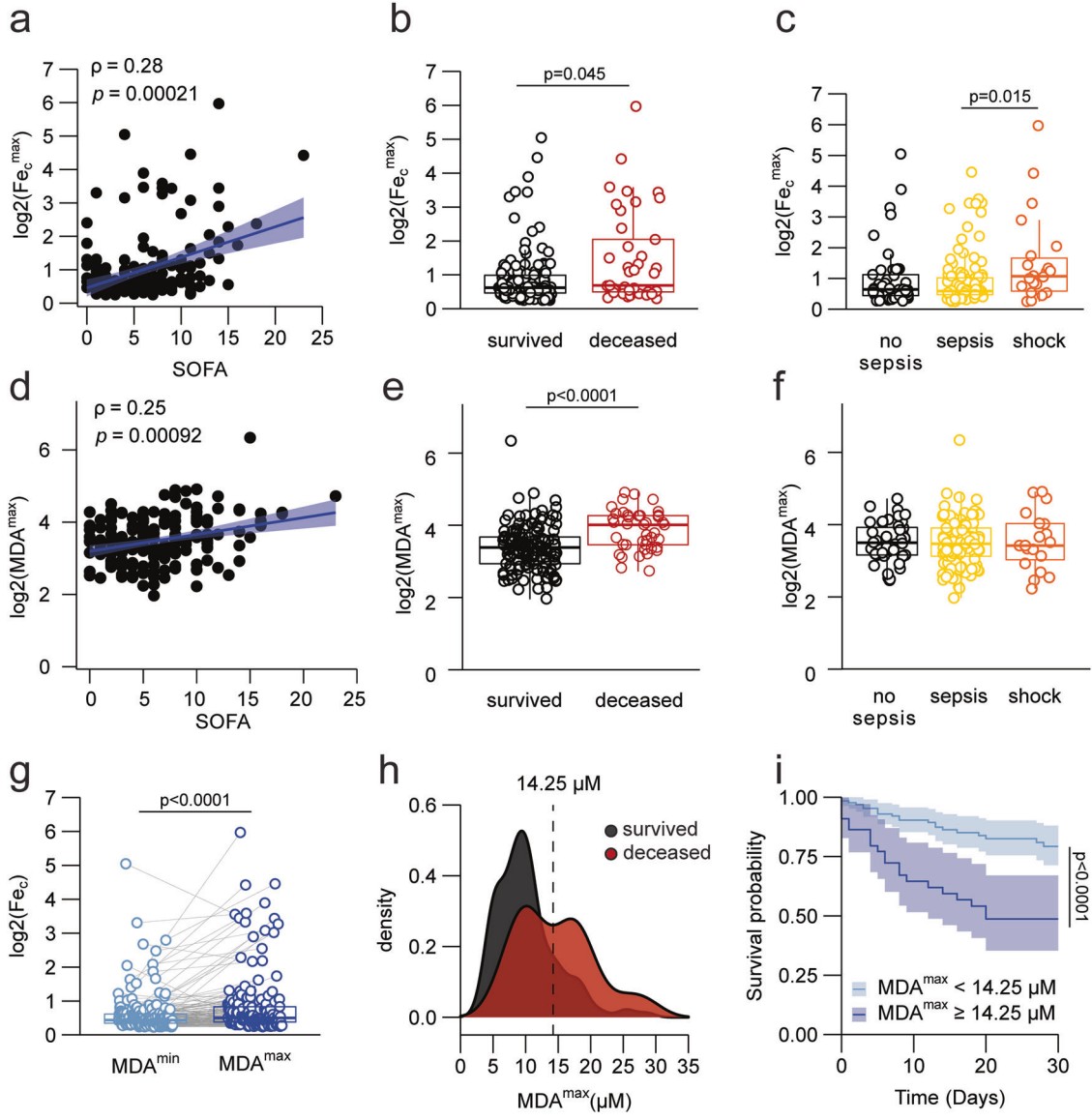

**Fig. 1 Maximum plasma MDA levels in critically ill patients are associated with plasma catalytic iron ($Fe_c$) levels, SOFA score, and mortality. a** Scatter plot showing the positive association between the log-transformed values for maximum catalytic iron ($Fe_c^{max}$) concentration per patient and the corresponding SOFA scores of that day. **b** Boxplots showing the log-transformed $Fe_c^{max}$ values of patients who passed away within 30 days and those who survived. **c** Boxplots showing the log-transformed $Fe_c^{max}$ values of all patients grouped by the presence of sepsis, septic shock, or non-septic MODS. **d** Scatter plot showing the positive association between the log-transformed values for maximum MDA ($MDA^{max}$) concentration per patient and the corresponding SOFA scores of that day. **e** Boxplots showing the log-transformed $MDA^{max}$ values of patients who passed away within 30 days and those who survived. **f** Boxplots showing the log-transformed $MDA^{max}$ values of all patients grouped by the presence of sepsis, septic shock, or non-septic MODS. **g** Boxplots showing the log-transformed $Fe_c$ values on the day of $MDA^{min}$ (light blue) and $MDA^{max}$ (dark blue) for each individual patient (gray lines). **h** Histogram representing the density distribution of $MDA^{max}$ values for both the patients who survived (black) versus those who died (red) within 30 days. **i** Survival curves representing the patients with values of $MDA^{max} < 14.25\,\mu M$ (light blue) and $MDA^{max} \geq 14.25\,\mu M$ (dark blue). Data were analyzed using Spearman's rank correlation coefficient for continuous variables (**a**, **d**), two-sided Wilcoxon–Mann–Whitney (**b**, **e**), Kruskal–Wallis (omnibus), and two-sided Wilcoxon–Mann–Whitney (pairwise) test (**c**, **f**), Paired two-sided Wilcoxon signed-rank test (**g**) and two-sided Log-Rank test (**i**). The full range of observations are plotted on all the boxplots, where the bottom, center, and top of the bounding box represent respectively the 25th, 50th, and 75th percentiles. The whiskers extend to a maximum of 1.5 times the interquartile range (IQR = Q3 − Q1) beyond the bounding box. $N = 176$ for all graphs. Source data are provided as a Source Data file.

ileum, and skeletal muscle tissue, as well as in plasma (Fig. 2c and Supplementary Fig. 3g). In addition, an increased number of dead cells as a function of time was detected in kidney, liver, and ileum tissue, reflected by a terminal deoxynucleotidyl transferase dUTP nick end labeling (TUNEL) staining (Fig. 2d–g). Hematologic analysis revealed leukocytosis, in particular neutrophilia and lymphopenia, which is also typically observed in patients with acute iron overload[29] (Supplementary Fig. 4c, d). Lastly, plasma

analysis of a panel of cytokines and chemokines displayed elevated levels of interleukin (IL)-6 upon acute iron overload (Supplementary Fig. 4e), likely representing a compensatory mechanism to inhibit intestinal iron uptake through hepcidin upregulation[30].

Increased levels of iron and consequent MDA pointed to the fact that ferroptosis rather than other modes of cell death is primarily responsible for the organ damage. Indeed, mice

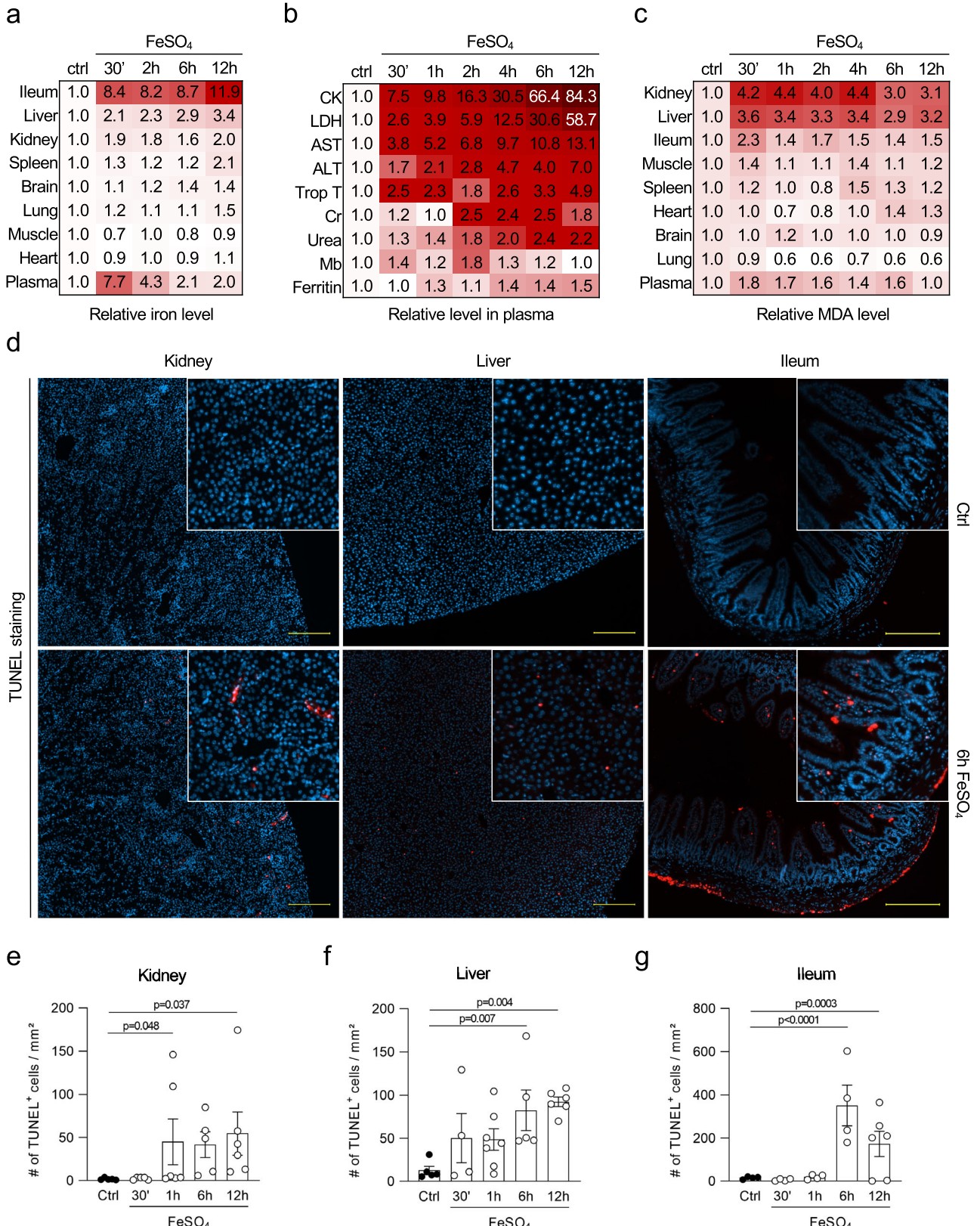

expressing a kinase-dead variant of receptor-interacting protein kinase 1 (RIPK1; Ripk1[ki/ki]) in which RIPK1 kinase-dependent apoptosis and necroptosis is blocked[31], showed no protection against acute iron overload (Fig. 3a–d). Several modes of regulated necrosis mediated by RIPK3, Poly (ADP-Ribose) polymerase 1 (PARP1) and Cyclophilin D (CYPD, encoded by

the ppif gene) have been reported to contribute to renal ischemia-reperfusion injury and/or consequent lung remote injury[32–34]. Upon acute iron overload, mice deficient in RIPK3, CYPD, and PARP1 (Ripk3[−/−]; Ppif[−/−]; Parp1[−/−]) only showed a mild drop in some plasma injury biomarkers compared to the wild type (WT) littermates (Fig. 3e–i). However, the reduction in organ

**Fig. 2 Acute iron overload in mice causes multiorgan failure due to excessive lipid peroxidation. a** Heatmap representing the relative iron levels in various organs after acute iron overload as a function of time. The combined results of three independent experiments are shown (total n = 5-6/ timepoint). **b** Heatmap representing the relative increase in creatine kinase (CK), lactate dehydrogenase (LDH), aspartate aminotransferase (AST), alanine aminotransferase (ALT), troponin T (Trop T), creatinine (Cr), urea, myoglobin (Mb), and ferritin after acute iron overload as a function of time in plasma. The combined results of minimum three independent experiments are shown (n = 6-13/time point). **c** Heatmap representing the relative MDA levels after acute iron overload as a function of time. The combined results of three independent experiments are shown (total n = 5-8/time point). **d** Immunohistochemical staining for cell death (TUNEL) in kidney, liver, and ileum 6 h after acute iron overload. Fluorescent photomicrographs representative for the outcome of three independent experiments (total n/time point described in panel **e**) are shown. Scale bar represents 200 μm. **e-g** Quantification of immunohistochemical staining for TUNEL in kidney (total n/condition from left to right is as follows: n = 5, 5, 6, 5, 6), liver (total n/condition from left to right is as follows: n = 5, 4, 7, 5, 6) and ileum (total n/condition from left to right is as follows: n = 4, 4, 4, 4, 6) sections after acute iron overload in function of time. The combined results of three independent experiments are shown. Heatmaps (**a-c**) show relative increase as color in two dimensions ranging from white (minimum) to red (maximum). Data were analyzed using one-way ANOVA followed by pairwise T-testing after fitting of a log-linear regression model (**e-g**). Means and SEM are represented. Source data are provided as a Source Data file.

damage was stronger when this mouse line was combined with overexpression of glutathione peroxidase 4 (GPX4) (GPX4$^{Tg/+}$; Fig. 3e–i), which inhibits ferroptosis by reducing phospholipid-hydroperoxides to their alcohol form[35]. This protective effect of GPX4 was confirmed in mice solely overexpressing GPX4 as well (GPX4$^{Tg/+}$; Fig. 3j–n). As a reverse strategy, we used mice that express a catalytically inactive form of GPX4 (cysteine-variant; Gpx4$^{fl/cys}$ R26CreERT2$^{Tg/+}$), referred to as ferroptosis sentinel mice[36]. Due to the inferior reductive capacity of this cysteine-variant to reduce phospholipid-hydroperoxides, these mice are sensitized to ferroptosis[36]. When subjected to acute iron overload, they showed a strong sensitization with significantly higher levels of plasma injury biomarkers compared to their littermate controls (Fig. 4a–d). Finally, we used a dietary approach by feeding the mice for 6 weeks with synthetic diets containing different amounts of vitamin E (dl-α-tocopheryl acetate), as a natural lipophilic radical trap inhibiting ferroptosis[37,38]. A high dietary dose of vitamin E reduced the plasma injury biomarkers after iron overload, while a near to deficient vitamin E diet strongly sensitized with sudden death as a result (Fig. 4e–h). These findings highlight ferroptosis as a key detrimental factor in iron overload-induced MODS.

**Ferroptosis targeting is a life-saving strategy to protect against experimental (multi)organ dysfunction.** We previously developed several novel Fer-analogs with improved stability, efficacy, and solubility[39,40], to allow in vivo targeting of ferroptosis. First, we performed an extended in vivo PK study of a selected highly soluble candidate lead ferroptosis inhibitor (UAMC-3203) in mice and rats after intravenous bolus administration (Supplementary Fig. 5a, b). The plasma concentration-time profile was best described using a 2-compartment model. A terminal half-life (t$_{1/2}$) of around 3–4 h in mice and 4–6 h in rats was determined for plasma and most tissues (Supplementary Fig. 5c, d). The median blood to plasma ratio in mice was 0.89 (Supplementary Fig. 5c), indicating minimal binding to blood cells. UAMC-3203 showed an extensive tissue distribution with tissue-to-plasma ratios ranging from 10.5 to 219 in mice and 12.5 to 114 in rats. In spinal fluid and brain, UAMC-3203 was hardly detectable, implying no or minor crossing of the blood-brain barrier. Thus, UAMC-3203 is a unique, highly soluble Fer-analog in 0.9% NaCl with favorable in vivo PK properties in mice and rats.

Next, to determine the efficacy of UAMC-3203 versus Fer1 and Lip1[41] in blocking ferroptosis in vivo, we generated inducible renal tubular epithelial (Gpx4$^{RTEKO}$) and hepatocyte specific Gpx4-deficient mice (Gpx4$^{HEPKO}$) (Fig. 5a and Supplementary Fig. 6a, b). Upon tamoxifen (TAM) application, Gpx4$^{RTEKO}$ and Gpx4$^{HEPKO}$ mice developed ferroptosis-driven acute kidney and liver dysfunction respectively, and consequently died. Six days following the last TAM injection Gpx4$^{RTEKO}$ mice showed an

increase in Cr and urea accompanied by extensive necrosis of the proximal tubules (Fig. 5b and Supplementary Fig. 6c). In particular, atypical cellular debris in the form of PAS-positive granules was observed, whereas the glomeruli appeared with dilated bowman's spaces (Supplementary Fig. 6c). Daily injection of UAMC-3203 or Lip1 following TAM treatment in Gpx4$^{RTEKO}$ mice could significantly delay death, while Fer1 provided no protection (Fig. 5c). In renal ischemia-reperfusion injury, UAMC-3203 also outperformed Fer1 as measured by the attenuation of tubular damage in the kidney (Supplementary Fig. 6e–g). In the case of ferroptosis-driven acute liver injury, Gpx4$^{HEPKO}$ mice showed highly elevated ALT, AST, and LDH levels concomitant with severe cell death and morphological liver tissue changes (Fig. 5d and Supplementary Fig. 6d, h) when sacrificing the mice upon a drop in body temperature. Tissue damage was characterized by enlarged nuclei, chromatin aberrations, and paling of both the hepatocellular nuclei and cytoplasm, likely reflecting death cell corpses. In the centrilobular region, mild inflammatory infiltrates were detected (Supplementary Fig. 6d). For Gpx4$^{HEPKO}$ mice, UAMC-3203 and Lip1 treatment showed a significantly stronger protection against TAM-induced acute liver dysfunction, plasma lipid peroxidation, and subsequent death than Fer1 (Fig. 5e–g), with almost normalized liver plasma injury biomarkers by day 21 when the mice were sacrificed (shown for UAMC-3203 Supplementary Fig. 6i).

Based on the favorable in vivo PK profile of UAMC-3203 as well as its efficacy, we then analyzed its effectiveness to block iron overload induced multiorgan dysfunction compared to Fer1 and Lip1. UAMC-3203 proved to be highly superior to Fer1 and slightly better than Lip1 based on the level of reduction in plasma injury biomarkers LDH, CK, AST and ALT, and body temperature (Fig. 6a–e), while the natural lipophilic radical trap α-tocopherol (α-toc) was ineffective in suppressing iron overload-induced injury, both at an equimolar concentration as well as in excess (Fig. 6f–j). The superior in vivo dampening of lipid peroxidation by UAMC-3203 compared to Fer1 was also reflected by the tissue MDA levels and a flow cytometric analysis of liver and kidney single-cell suspensions (Fig. 7a–c and Supplementary Fig. 7). To validate the efficacy of UAMC-3203 in a different model of MODS, it was tested in mice subjected to intravascular hemolysis caused by the interaction of phenylhydrazine (PHZ) with red blood cells[42]. UAMC-3203 treatment led to a significant but rather mild reduction of plasma LDH in this model (Supplementary Fig. 8a). Since UAMC-3203 does not accumulate in the red blood cells, its direct effect on hemolysis is limited (Supplementary Fig. 8b). Interestingly, treatment with UAMC-3203 had no effect on the plasma injury biomarkers in TNF-induced systemic inflammatory response syndrome (Supplementary Fig. 9a–d) or CLP induced septic shock (Supplementary Fig. 9e–h). Similarly, mice overexpressing GPX4 showed no or even slightly decreased

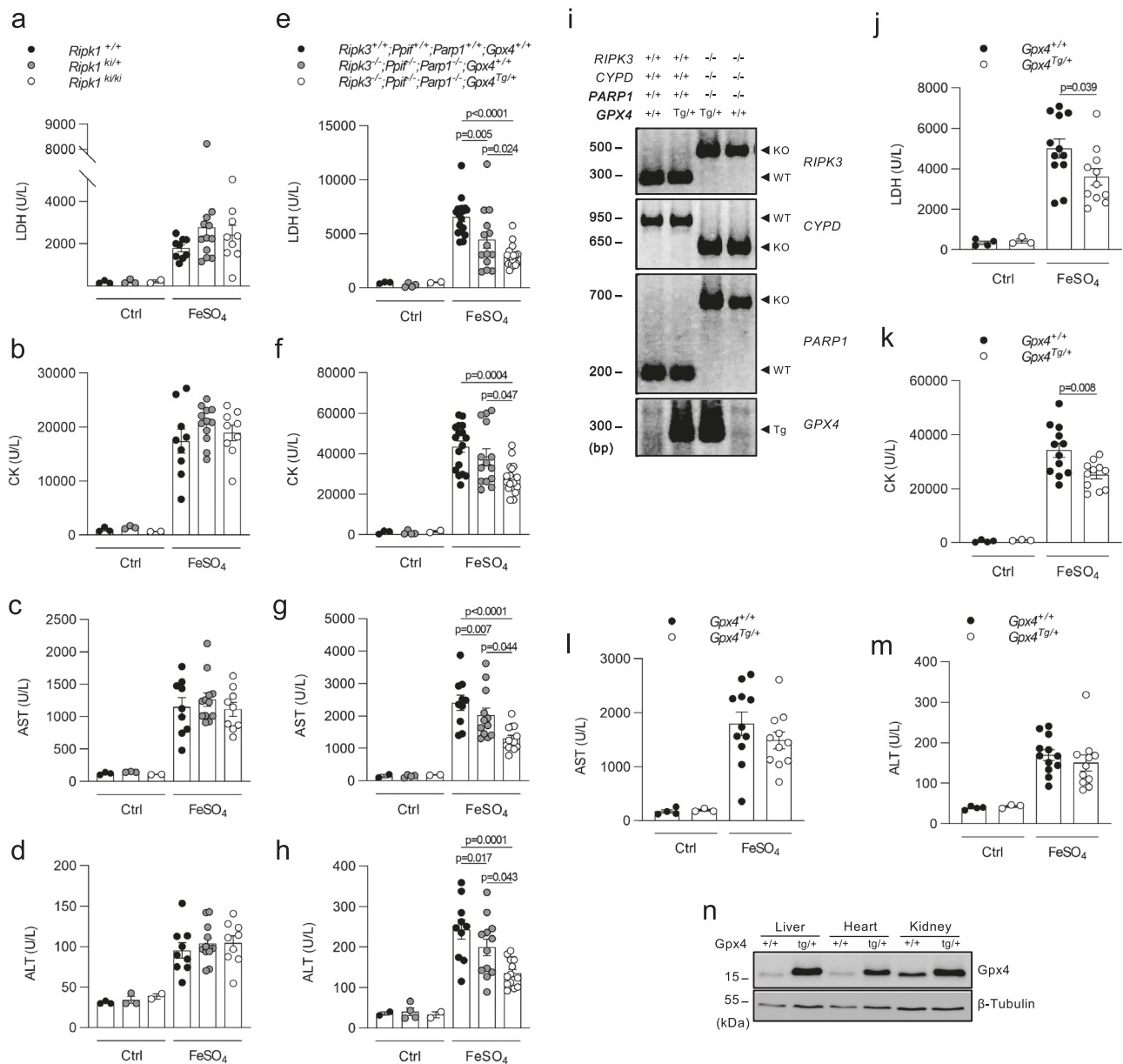

**Fig. 3 RIPK1, RIPK3, CypD, and PARP1 are not critically involved in iron overload-induced injury. a–d** Plasma levels of LDH, CK, AST, and ALT 2 h after iron overload for RIPK1 kinase-dead mice (*Ripk1*^ki/ki^). The combined results of minimum 2 independent experiments are shown (with total n/condition from left to right as follows: $n = 3, 3, 2, 9, 12, 9$). **e–h** Plasma levels of LDH, CK, AST, and ALT 2 h after iron overload for mice triple-deficient in RIPK3, CypD, and PARP1 overexpressing GPX4 (*Ripk3*^−/−^; *Ppif*^−/−^; *Parp*^−/−^; *GPX4*^Tg/+^). The combined results of minimum two independent experiments are shown (with total n/condition for LDH and CK from left to right as follows: $n = 3, 4, 2, 17, 14, 18$ and total n/condition for AST and ALT from left to right as follows: $n = 2, 4, 2, 10, 12, 13$). **i** PCR reaction illustrating the deficiency of Ripk3, Ppif, and Parp1 with or without transgenic expression of Gpx4 in the *Ripk3*^−/−^; *Ppif*^−/−^; *Parp*^−/−^*Gpx4*^Tg/+^ mouse line. The gel shown is representative of the outcome of minimum three independent litters. **j–m** Plasma levels of LDH, CK, AST, and ALT 2 h after iron overload for GPX4 overexpressing mice (*Gpx4*^Tg/+^). The combined results of minimum 2 independent experiments are shown (with total n/condition from left to right as follows: $n = 4, 3, 12, 11$ for LDH, CK, ALT; $n = 4, 3, 11, 11$ for AST). **n** Western blot illustrating the overexpression of GPX4 in liver, heart, and kidney tissue of *Gpx4*^Tg/+^ mice. The blot shown is representative of two independent experiments performed on liver tissue (1 experiment on heart and kidney tissue). These experiments were performed on a synthetic diet containing 50 mg/kg vitamin E. Data were analyzed using one-way ANOVA followed by pairwise *T*-testing (**a–h**) and unpaired, two-tailed *T*-testing (**j–m**). Means and SEMs are represented. Source data are provided as a Source Data file.

survival after CLP- and lipopolysaccharide (LPS)-induced lethal shock respectively (Supplementary Fig. 9i, j). Noteworthy, reduced levels of plasma iron were detected after TNF or CLP challenge (Supplementary Fig. 9k, l), presumably as a protective strategy to limit microbial growth by reducing their iron uptake[30,43].

Considering the high mortality in critically ill patients with MODS, we finally analyzed the potency of UAMC-3203 to

protect against iron-dependent multiorgan dysfunction induced death. When dissolving the compounds in 0.9% NaCl containing 2% DMSO, only UAMC-3203 showed a small beneficial effect over vehicle (Fig. 7d). However, owing to its high solubility, UAMC-3203, but not Fer1 or Lip1 (Supplementary Fig. 5b), can be administered in 0.9% NaCl without DMSO. Using this physiologic solvent, UAMC-3203 protected 60% of mice against

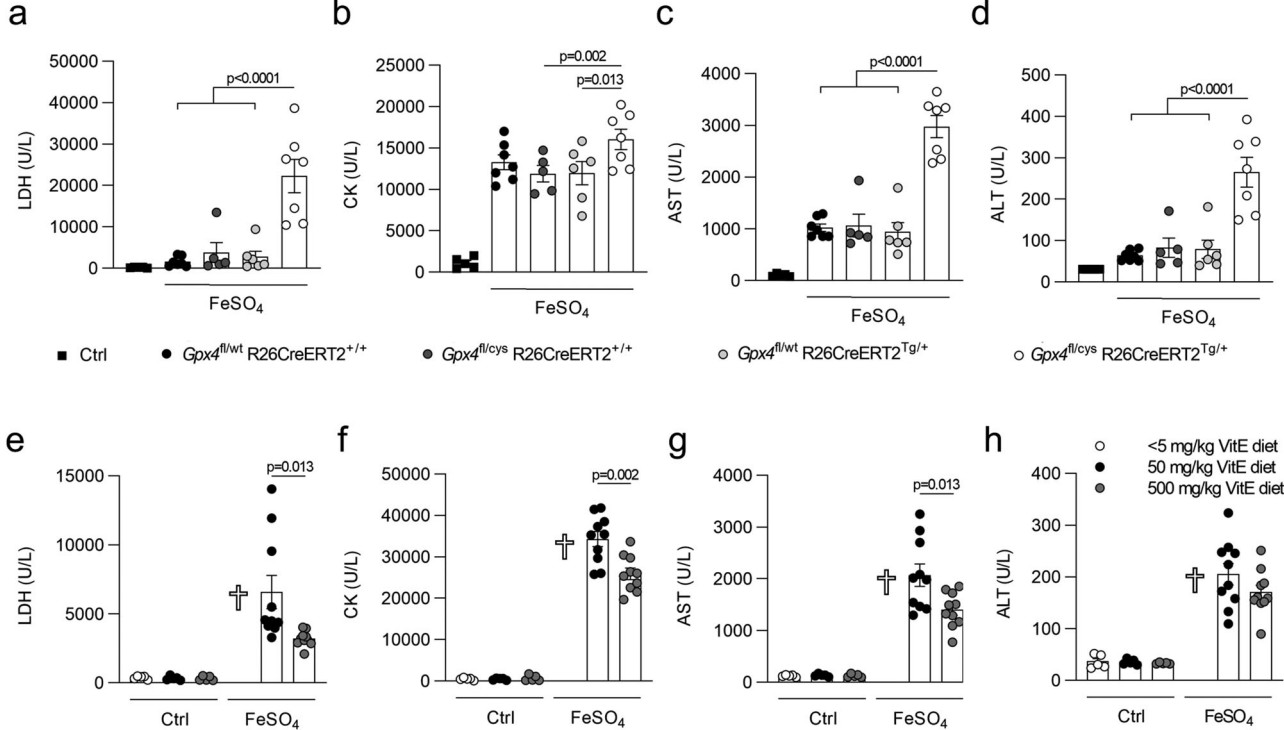

**Fig. 4 High vitamin E diet reduces injury by iron overload, while ferroptosis sentinel mice expressing the cysteine mutant of GPX4 show a strong exacerbation. a–d** Plasma levels of LDH, CK, AST, and ALT 2 h after iron overload for mice expressing the GPX4 cysteine mutant. The combined results of minimum two independent experiments are shown (with total n/condition from left to right as follows: n = 5, 7, 5, 6, 7). **e–h** Plasma levels of LDH, CK, AST, and ALT 2 h after iron overload for mice receiving different dietary levels of vitamin E (<5/50/500 mg/kg). The combined results of two independent experiments are shown (with total n/condition = 5 for the ctrl groups resp. 10 for the FeSO4 treated groups). The cross symbol represents death of the mice. Data were analyzed using one-way ANOVA followed by pairwise T-testing (**a–d**) and unpaired, two-tailed T-testing (**e–h**). Means and SEMs are represented. Source data are provided as a Source Data file.

the severe model of iron overload-induced lethality (Fig. 7e). Thus, this unique solubility and efficacy makes UAMC-3203 a superior life-saving ferroptosis inhibitor, which shows strong protection against multiorgan dysfunction.

## Discussion

MODS is a complex, multifactorial pathology of which the underlying mechanism is not well understood. Consequently, there is a lack of predictive biomarkers to be used as prognostic tools or to guide therapeutic decision making[1]. We found that the severity of multiorgan dysfunction and the probability of death among critically ill patients is associated with plasma $Fe_c$ levels and excessive lipid peroxidation. These results are in line with previous observations in critically ill patients indicating an association between either plasma $Fe_c$ or MDA levels[44,45] and worsening of the disease. However, by analyzing both parameters in the same cohort, we showed a correlation between plasma $Fe_c$ and MDA levels alike, suggesting a common detrimental molecular mechanism. Based on elevated levels of lipid peroxidation, a subpopulation was identified to be at considerably higher risk of death, making MDA measurements a promising prognostic tool.

While critical illness displays a high level of complexity, the association of elevated iron and lipid peroxidation levels with poor outcome suggests that ferroptosis can be a detrimental factor during the onset and progression of MODS. Indeed, we find that excessive iron can induce multiorgan dysfunction in mice, which is dominantly driven by ferroptosis through excessive lipid peroxidation-induced injury. Fer-analog UAMC-3203 is a potent inhibitor of ferroptosis with a superior PK profile compared to

Fer1[40]. Here, we show that UAMC-3203 outperforms Fer1 and Lip1 in vivo in its ability to reduce both single and multiple organ dysfunction, and prevent death.

As for any experimental model, there are limitations to the different mouse models used in our study. Firstly, the genetic models in which ferroptosis is induced by depletion of GPX4 in liver or kidney tissue conceptually show that UAMC-3203 is a more potent ferroptosis inhibitor in liver compared to kidney. These models are however not suitable to study ferroptosis in the context of multi-organ injury and are not representative for critically ill patients in terms of disease etiology.

Secondly, although the levels of $Fe_c$ in critically ill patients associate with organ dysfunction and death, it remains unclear to what extent experimental iron overloading accurately reflects the situation of MODS patients in the ICU. Surely the model mimics the organ injury observed in patients with iron intoxication. Thirdly, PHZ was used to pharmacologically induce intravascular hemolysis and secondary MODS, which is only occasionally observed in MODS patients. Additionally, PHZ causes the formation of hydrogen peroxide molecules[46], the toxicity of which cannot normally be inhibited by lipophilic radical traps[36,41], potentially explaining the modest effect of UAMC-3203. Nonetheless, the robust results obtained both in human subjects and in vivo highlight the need for further exploration of ferroptosis its role in MODS.

Perhaps surprisingly, patients suffering from sepsis did not exhibit higher plasma $Fe_c$ or MDA levels than non-septic MODS patients. Most prior studies investigating catalytic iron and lipid peroxidation in the context of critical illness either did not make the distinction between septic and non-septic patients on the ICU[6,44], or specifically selected only septic patients which were

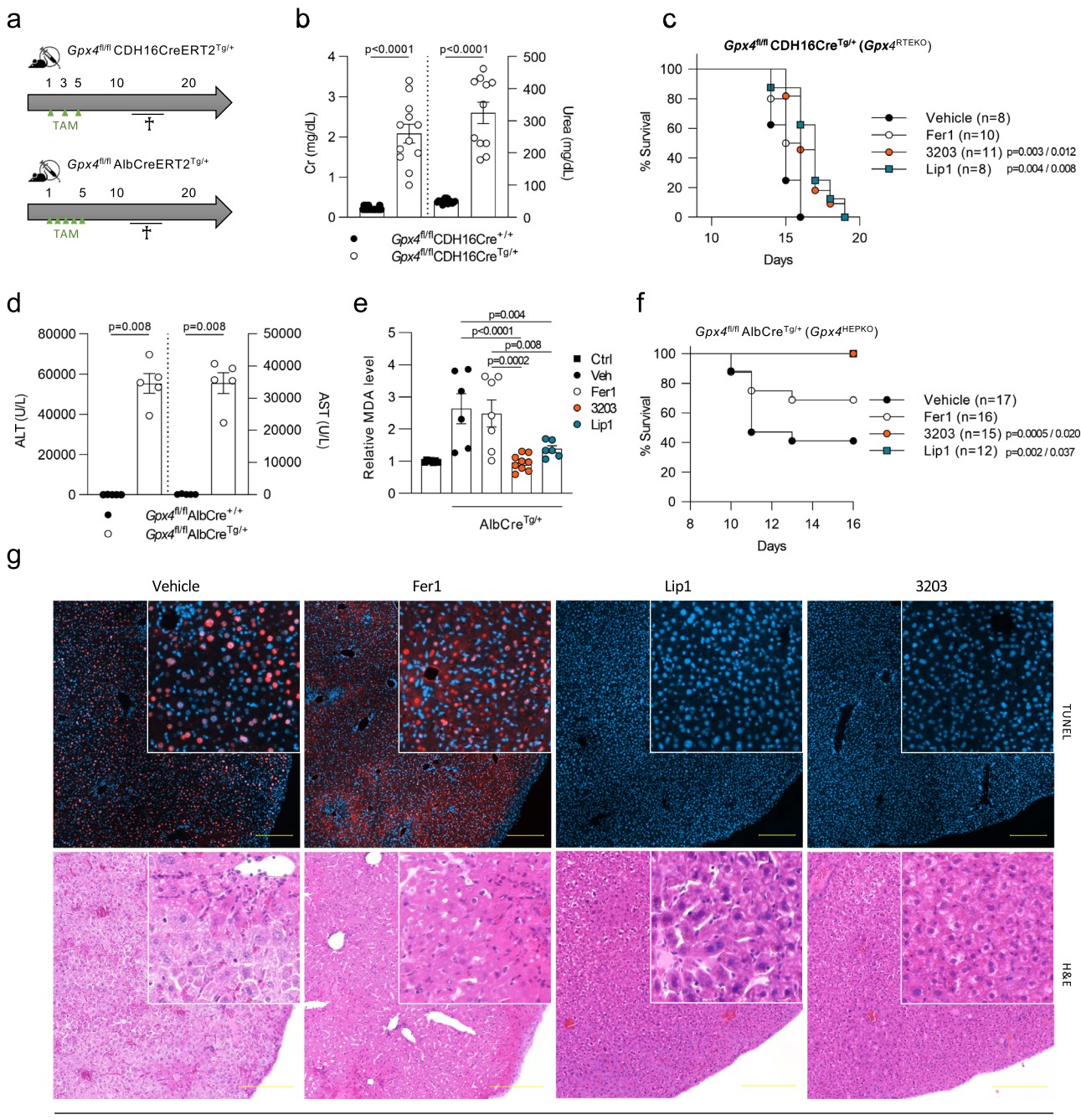

compared to healthy controls[45,47] or among themselves based on mortality[48]. One small study corroborates our findings with both uninfected critically ill patients and septic patients having comparably elevated MDA levels in relation to healthy volunteers[49]. Similarly, Leaf et al. observed no difference in the magnitude of association between $Fe_c$ and mortality on the basis of sepsis or shock[5]. When a small set of septic patients was divided based on number of secondary organs failing, plasma MDA levels were significantly higher in patients presenting failure of three or more secondary organs[50]. This implies that the aspect of multi-organ dysfunction rather than the presence of infection is linked to the dysregulation of iron homeostasis and consequent lipid peroxidation. Indeed, our results show that both a pharmacological and genetic approach to inhibit ferroptosis could not improve the outcome in classical animal models of sepsis and septic shock. We

therefore envision that ferroptosis inhibition could be a promising therapeutic strategy for non-septic patients with MODS (e.g. trauma), while for septic shock patients with MODS a combination treatment might be needed to control systemic inflammation as well, as we previously reported *viz.* simultaneous neutralization of IL-1 and -18[51].

In conclusion, these results should encourage further research into the targeting of ferroptosis as a novel therapeutic strategy for patients with either acute single or multiorgan dysfunction, which remains one of the major life-threatening conditions in critical illness. Plasma MDA levels hold prognostic value and can allow patient stratification for future treatment regimens. The Fer-analog UAMC-3203, or a new derivative thereof, could be considered as a candidate lead ferroptosis inhibitor for clinical translation.

**Fig. 5 The highly soluble Fer-analog UAMC-3203 is a potent inhibitor of ferroptosis in kidney and liver. a** Schematic representation of tamoxifen (TAM) ip injection regime (green triangles) of $Gpx4^{fl/fl}$ CDH16CreERT2$^{Tg/+}$ mice or $Gpx4^{fl/fl}$ AlbCreERT2$^{Tg/+}$ mice. **b** Plasma creatinine (Cr) and urea levels of $Gpx4^{fl/fl}$ CDH16CreERT2$^{Tg/+}$ mice sacrificed 6 days after TAM administration. The combined results of minimum two independent experiments are shown (total $n = 17$ for CreERT2$^{+/+}$ mice, total $n = 11$ resp. 12 for urea resp. creatinine of CreERT2$^{Tg/+}$ mice). **c** Survival curve of $Gpx4^{fl/fl}$ CDH16CreERT2$^{Tg/+}$ mice treated daily with vehicle (2% DMSO), Fer1, Lip1, or compound UAMC-3203 starting two days prior to TAM-mediated Cre activation. The combined results of minimum two independent experiments are shown. **d** Plasma ALT and AST levels of $Gpx4^{fl/fl}$ AlbCreERT2$^{Tg/+}$ mice sacrificed once a human endpoint was reached. The combined results of two independent experiments are shown (total $n = 5$/condition). **e** Relative plasma MDA levels of WT littermates (ctrl; total $n = 7$) or $Gpx4^{fl/fl}$AlbCreERT2$^{Tg/+}$ mice treated daily with vehicle (2% DMSO; total $n = 6$), Fer1 (total $n = 7$), Lip1 (total $n = 6$), or compound UAMC-3203 (total $n = 9$) starting 1 day after TAM-mediated Cre activation, sacrificed once a human endpoint was reached or at the end of the experiment. The combined results of minimum two independent experiments are shown. **f** Survival curve of $Gpx4^{fl/fl}$AlbCreERT2$^{Tg/+}$ mice treated daily with vehicle (2% DMSO), Fer1, Lip1, or compound UAMC-3203 starting 1 day after TAM-mediated Cre activation. The combined results of minimum three independent experiments are shown. **g** (Immuno)histochemical TUNEL and H&E staining of liver tissue of $Gpx4^{fl/fl}$ AlbCreERT2$^{Tg/+}$ mice treated daily with vehicle (2% DMSO; total $n = 4$), Fer1 (total $n = 3$), Lip1 (total $n = 6$) or compound UAMC-3203 (total $n = 9$), sacrificed once a human endpoint was reached or at the end of the experiment. Photomicrographs representative for the outcome of minimum two independent experiments are shown. Scale bar represents 200 μm. Data were analyzed using unpaired, two-tailed $T$-testing (**b**, **d**), one-way ANOVA followed by pairwise $T$-testing, (**e**) and the Mantel–Cox test (**c**, **f**). Means and SEMs are represented; $P$ in **c**, **f** represents the statistical significance compared to vehicle (left) resp. Fer1 (right). Source data are provided as a Source Data file.

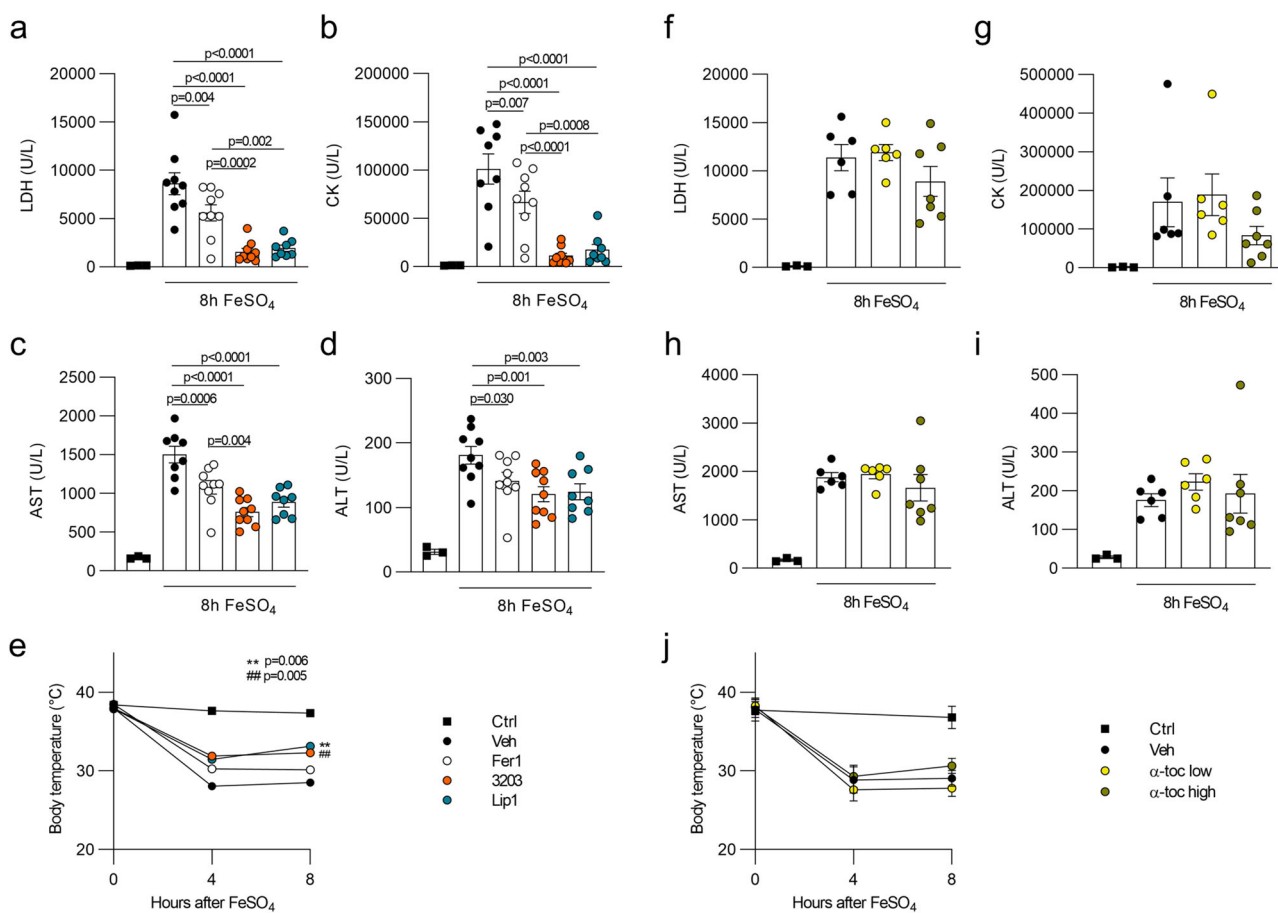

**Fig. 6 Fer-analog UAMC-3203 is superior in reducing organ injury caused by iron overload. a–e** Plasma levels of LDH, CK, AST, and ALT as well as body temperature for control mice (total $n = 3$) and mice subjected to acute iron overload 8 h after treatment with vehicle (2% DMSO; total $n = 9$ for LDH, ALT, body temperature resp. 8 for CK, AST), Fer1 (total $n = 9$ for LDH, CK, AST, ALT; total $n = 10$ for body temperature), Lip1 (total $n = 8$) or compound UAMC-3203 (total $n = 9$ for LDH, CK, AST, ALT; total $n = 10$ for body temperature). The combined results of minimum two independent experiments are shown. **f–j** Plasma levels of LDH, CK, AST and ALT as well as body temperature for control mice (total $n = 3$) and mice subjected to acute iron overload 8 h after treatment with vehicle (corn oil; total $n = 6$) or α-tocopherol in low (8.6 mg/kg; total $n = 6$) or high (400 mg/kg; total $n = 7$) concentration. The combined results of minimum two independent experiments are shown. Data were analyzed using one-way ANOVA followed by pairwise $T$-testing (**a–d**, **f–i**), and as repeated measurements followed by an approximate $F$-test for differences in treatment averaged over and across time (**e**, **j**). Means and SEMs are represented. Statistical differences shown for body temperature are compared to vehicle treatment averaged over time (**e**). When evaluated across time, $P < 0.05$ for UAMC-3203 and Fer1 compared to vehicle, and for Lip1 compared to Fer1 (**e**). Source data are provided as a Source Data file.

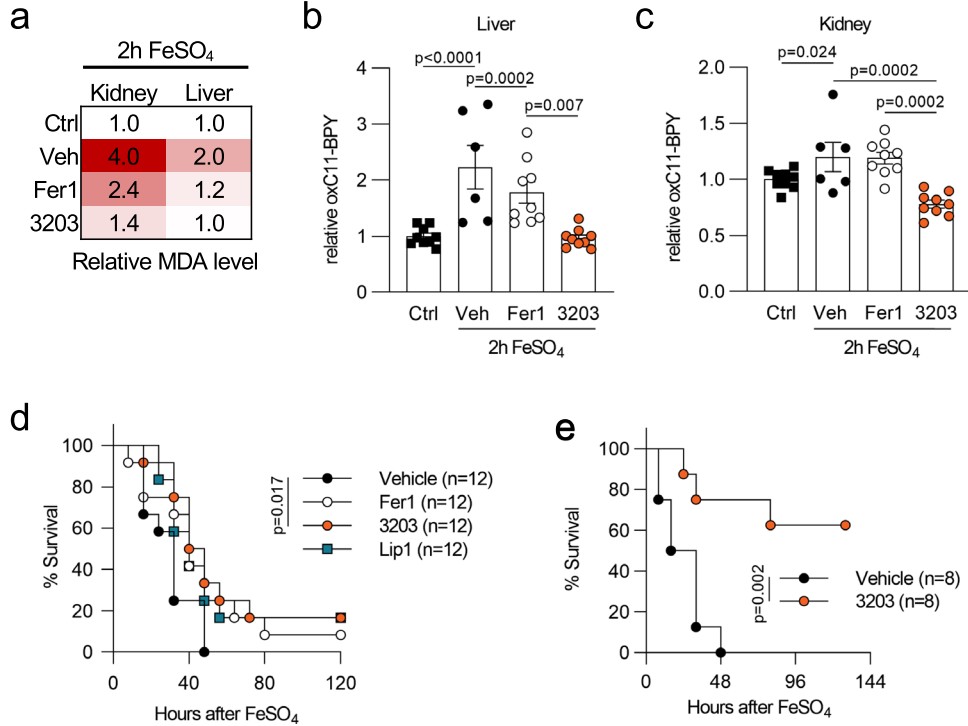

**Fig. 7 Fer-analog UAMC-3203 is superior in reducing lipid peroxidation and death caused by iron overload. a** Heatmap representing the relative MDA levels for kidney (total $n = 7$ for ctrl, Fer1, UAMC-3203; total $n = 5$ for vehicle group) and liver (total $n = 7$/condition) 2 h after acute iron overload for mice treated with vehicle (2% DMSO), Fer1 or compound UAMC-3203. The combined results of two independent experiments are shown. **b, c** Relative measure of lipid peroxidation in liver and kidney tissue detected by flow cytometry in the form of oxidized C11-BODIPY staining 30 min after acute iron overload for mice treated with vehicle (2% DMSO), Fer1, or compound UAMC-3203. The combined results of three independent experiments are shown (total $n = 9$ for ctrl, Fer1, UAMC-3203; total $n = 6$ for vehicle group). **d** Survival curve after acute iron overload for mice treated three times daily with vehicle (2% DMSO), Fer1, Lip1, or UAMC-3203. The combined results of two independent experiments are shown. **e** Survival curve after acute iron overload for mice treated three times daily with vehicle (0.9% NaCl) or UAMC-3203. The combined results of two independent experiments are shown. Heatmap (**a**) shows relative increase as color in two dimensions ranging from white (minimum) to red (maximum). Data were analyzed using one-way ANOVA followed by pairwise $T$-testing (**b**, **c**), and via the Mantel–Cox test (**d**, **e**). Means and SEMs are represented. Source data are provided as a Source Data file.

## Methods

**ICU patient blood collection**. The prospective cohort study in adult critically ill patients was approved by the Ethical Committee of the Ghent University Hospital (Belgian registration number of the study: B670201213147), and conducted in accordance with the declaration of Helsinki and in compliance with the Good Clinical Practice Guidelines. All patients or their legally authorized representatives provided written informed consent. The study consisted of 176 participants, as previously described[26]. Briefly, blood samples were collected daily for seven consecutive days. The first blood samples were collected at enrollment (day 1) and subsequent sampling took place daily at 6 am.

**Mice**. All experiments were approved by the animal ethics committee of Ghent University (EC2013-045, EC2014-065, EC2016-023, EC2017-055, EC2017-097, EC2017-098, EC2017-099, EC2019-013, EC2019-078, EC2020-082, EC2020-106), Antwerp University (2019-19), or by ethic committees and local authorities in Dresden (TVA 07/2021 Landesdirektion Sachsen) and conducted according to institutional, national and European animal regulations. Mice were bred and housed under SPF conditions in individually ventilated cages, in conventional, temperature-controlled (21 °C) animal facilities with a 14/10-h light/dark cycle and humidity of 55%. All mice used in the experiments were between the age of 8–13 weeks. Experiments were performed with C57BL/6 N mice ordered from Janvier Labs unless mentioned otherwise. *Ripk1*ki mice were purchased from GSK[31]. Double deficient *Ripk3*−/−;*Ppif*−/− mice were kindly provided by A. Linkermann[32] and crossed with *Parp1*−/− mice received from F. Dantzer[52] and *Gpx4*Tg/+ mice from Q. Ran[53]. Both the *Gpx4*fl/cys R26CreERT2Tg/+ and the *Gpx4*fl/fl mice were kindly provided by M. Conrad[35,36], while the AlbCreERT2Tg/+ and CDH16CreERT2Tg/+ mice were received from D. Metzger[54] and D. Peters[55], respectively. The *Gpx4*fl/fl mice were crossed with the AlbCreERT2Tg/+ and CDH16CreERT2Tg/+ mice to obtain an inducible liver-specific resp. kidney-specific Gpx4 deficient mouse line. Genetically modified mice were always compared to their wild-type littermates. Water and all feeds used were provided ad libitum. Synthetic diet containing adapted levels of vitamin E (Ssniff) was given from the age of 4 weeks up until termination of the experiment.

**Genotyping**. The genotyping strategies used for the different mouse lines are described below. Ripk1ki mice were identified by use of PCR primers: 5′-CTC TGATTGCTTTATAGGACACAGCA and 5′-GTCTTCAGTGATGTCTTCCTCG TA, yielding a 473 bp wild-type DNA fragment and a 575 pb mutant DNA fragment. The Ripk3−/−; Ppif−/−; Parp1−/−; Gpx4Tg/+ and Gpx4Tg/+ mice created in house were genotyped with following PCR primers: 5′-GCCTGCCCATCAGCAAC TC, 5′- CCAGAGGCCACTTGTGTAGCG and 5′- CGCTTTAGAAGCCTTCA GGTTGAC for RIPK3, yielding a 300 bp wild-type DNA fragment and a 500 bp mutant DNA fragment; 5′-CTCTTCTGGGCAAGAATTGC, 5′- ATTGTGGTTGG TGAAGTCGCC and 5′- GGCTGCTAAAGCGCATGCTCC for Ppif, yielding a 950 bp wild-type DNA fragment and a 650 bp mutant DNA fragment; 5′- CTTG ATGGCCGGGAGCTGCTTCTTC, 5′-GGCCAGATGCGCCTGTCCAAGAAG and 5′- GGCGAGGATCTCGTCGTGACCCATG for Parp1, yielding a 200 bp wild-type DNA fragment and a 700 bp mutant DNA fragment; and 5′-CGTGG AACTGTGAGCTTTGTG and 5′- AAGGATCACAGAGCTGAGGCTG for Gpx4, yielding a 300 bp DNA fragment upon overexpression. Since it was impossible to distinguish between wild-type and point-mutated Gpx4 via PCR primer binding, the breeding strategy was organized so that Gpx4fl/cys R26CreERT2Tg/+ mice could be discriminated from their littermate controls via PCR primers binding both regular Gpx4 and Gpx4 flanked by flox regions, yielding a 180 bp DNA fragment for regular Gpx4 and a 240 bp DNA fragment when floxed: 5′- CGTGGAACT GTGAGCTTTGTG and 5′- AAGGATCACAGAGCTGAGGCTG. The same primers were used to identify floxed alleles in the Gpx4fl/fl AlbCreERT2Tg/+ and Gpx4fl/cys CDH16CreERT2Tg/+ mouse lines, for which the presence of Cre was evaluated with PCR primer 5′- GCCTGCATTACCGGTCGATGCAACGA and 5′- GTGGCAGATGGCGCGGCAACACCATT, yielding an 800 bp DNA fragment upon presence of Cre. Effective introduction of Gpx4 deficiency in liver resp. kidney tissue after tamoxifen administration was evaluated using PCR primers 5′- GTGTACCACGTAGGTACAGTGTCTGC and 5′- GGATCTAAGGATCAC AGAGCTGAG GCTGC, yielding a 500 bp DNA fragment upon Gpx4 excision.

**Acute iron overload model**. Mice treated with iron(II) sulfate heptahydrate received an intraperitoneal (i.p.) injection of 300 mg/kg body weight FeSO4.7H2O (Sigma; #F7002-250G-D) dissolved in sterile 0.9% sodium chloride (NaCl) with an

injection volume of 200 μL/20 g body weight. Rectal body temperature was monitored daily or more with an industrial electric thermometer (Comark Electronics, Norwich, UK; model 2001). Upon termination of the experiment, mice were anesthetized with isoflurane, blood was sampled in EDTA-coated tubes, and mice were sacrificed by cervical dislocation followed by dissection of the organs. When comparing Fer1, Lip1 and UAMC-3203 in this model, equimolar amounts of the compounds were dissolved in sterile 0.9% NaCl containing 2% dimethyl sulfoxide (DMSO,Sigma; #D-2650). In all but the survival experiments, a concentration of 2 mM with an injection volume of 200 μL/20 g body weight was administered corresponding to 12.35 mg/kg UAMC-3203, 6.8 mg/kg Lip1, and 5.2 mg/kg Fer1. In the survival experiment the compounds were administered three times daily at a concentration of 4 mM with an injection volume of 50 μL/20 g body weight. For experiments in which the mice were sacrificed at a time point of 2 h or earlier, the compounds were injected intraperitoneally 15 min prior to iron sulfate injection, while for experiments with 8 h of sacrifice, the compounds were administered 30 min after iron sulfate injection. For experiments in which the mice were sacrificed at a time point of 8 h or later, 0.4 mg/mL ibuprofen (Reckitt Benckiser; #2922607) was added to the drinking water 16 h prior to iron sulfate injection.

**Phenylhydrazine (PHZ) model**. Mice on a C57BL/6 N background bred in-house received an i.p. injection of 60 mg/kg PHZ (Merck; # 1072510100) diluted in endotoxin-free phosphate-buffered saline (PBS) with an injection volume of 100 μL/20 g body weight. UAMC-3203 dissolved in sterile 0.9% NaCl was injected i.p. at a concentration of 4 mM with an injection volume of 100 μL/20 g body weight 1 h prior to the administration of PHZ. Rectal body temperature was recorded with an industrial electric thermometer (Comark Electronics, Norwich, UK; model 2001). Twenty-four hours after PHZ treatment, mice were anesthetized with isoflurane, blood was sampled in EDTA-coated tubes, and mice were sacrificed by cervical dislocation.

**Tumor necrosis factor (TNF) shock model**. C57BL/6 J mice received an intravenous (i.v.) injection of 500 μg/kg body weight murine TNF (VIB Protein Service Facility; Ghent, Belgium) diluted in endotoxin-free PBS with an injection volume of 200 μL/20 g body weight. UAMC-3203 dissolved in sterile 0.9% NaCl containing 2% DMSO (Sigma; #D-2650) was injected i.p. at a concentration of 2 mM with an injection volume of 200 μL/20 g body weight 30 min prior to the administration of TNF. Rectal body temperature was recorded with an industrial electric thermometer (Comark Electronics, Norwich, UK; model 2001). Eight hours after TNF treatment, mice were anesthetized with isoflurane, blood was sampled in EDTA-coated tubes, and mice were sacrificed by cervical dislocation.

**Cecal ligation and puncture (CLP) sepsis model**. Two different procedures were used (mild or severe) when the mice were sacrificed after 24 h (mild model) and when a survival experiment was performed (severe model). Briefly, the mice were anesthetized using 2% isoflurane in oxygen. After hair removal and disinfection of the abdomen, a 10-mm midline laparotomy was performed, and the cecum exposed. Using 5–0 Ethicon Mersilk suture (Ethicon, Norderstedt, Germany), 50% of the cecum was ligated and subsequently perforated by a single through-and-through puncture with a 22 G needle (for mild CLP), or 100% of the cecum was ligated and subsequently perforated twice by a through-and-through puncture with a 20 G needle (for severe CLP). The abdomen was closed in two layers, using 5–0 sutures for the peritoneum and abdominal musculature, and wound clips for the skin. Following surgery, the animals were resuscitated with 1 ml prewarmed 0.9% saline administered subcutaneously. In addition, the mice subjected to the severe CLP procedure were treated intraperitoneally with broad-spectrum antibiotics (ciprofloxacin, 4 mg/kg, ampicillin, 20 mg/kg, metronidazole, 20 mg/kg, and vancomycin, 10 mg/kg together in 400 μl PBS) or vehicle (400 μl PBS) daily until day 10. On day 0, the mice received the antibiotic treatment 5 h after the CLP procedure. Sham-operated mice underwent the same procedure but without CLP. All animals were given preoperative and postoperative analgesia (0.4 mg/mL ibuprofen, Reckitt Benckiser; #2922607) in the drinking water, starting 24 h before until 48 h after surgery. In the mild model, UAMC-3203 dissolved in sterile 0.9% NaCl containing 2% DMSO (Sigma; #D-2650) was injected i.p. at a concentration of 2 mM with an injection volume of 200 μL/20 g body weight both 30 min and 8 h after surgery. Rectal body temperature was monitored daily with an industrial electric thermometer (Comark Electronics, Norwich, UK; model 2001). Upon termination of the experiment, mice were anesthetized with isoflurane, blood was sampled in EDTA-coated tubes, and mice were sacrificed by cervical dislocation.

**LPS sepsis model**. Mice were injected i.p. with 10 mg/kg body weight lipopolysaccharide (LPS) from Escherichia coli O111:B4 (Sigma-Aldrich, #2630) suspended in LPS-free PBS with an injection volume of 200 μL/20 g body weight. Rectal body temperature was monitored daily with an industrial electric thermometer (Comark Electronics, Norwich, UK; model 2001). Upon termination of the experiment, mice were sacrificed by cervical dislocation.

**Cre activation of the GPX4 floxed mice**. Disruption of the loxP-flanked Gpx4 allele upon Cre activation was induced by 3 ($Gpx4^{fl/fl}$ CDH16CreERT2$^{Tg/+}$; Gpx4$^{fl/cys}$ R26CreERT2$^{Tg/+}$) resp. 5 ($Gpx4^{fl/fl}$ AlbCreERT2$^{Tg/+}$) i.p. injections of 100 mg/kg

Tamoxifen (Sigma-Aldrich T-5648) dissolved to 20 mg/ml in corn oil containing 10% ethanol. Fer1, UAMC-3203 and vehicle were administered daily via i.p. injections (in sterile 0.9% NaCl containing 2% DMSO, Sigma; #D-2650) at a concentration of 2 mM with an injection volume of 200 μL/20 g body weight starting 1 day after the final Tamoxifen injection for the $Gpx4^{fl/fl}$ AlbCreERT2$^{Tg/+}$ line and 2 days prior to the Tamoxifen injections for the $Gpx4^{fl/fl}$ CDH16CreERT2$^{Tg/+}$ line. The mice were monitored daily for weight, temperature, and signs of morbidity, and sacrificed prior to termination of the experiment by cervical dislocation if a human endpoint was reached.

**Bilateral kidney ischemia reperfusion injury**. Bilateral kidney ischemia/ reperfusion injury (IRI) was performed as described previously[56]. Mice received UAMC-3203, Fer1 (at a concentration of 2 mM with an injection volume of 200 μL/20 g body weight) or matched vehicle (2% DMSO in sterile 0.9% NaCl) 15 min prior to surgery by i.p. injection. 5 min prior to anesthesia, all mice received 0.1 μg/g body weight buprenorphine-HCl intraperitoneally for analgesia. Anesthesia was induced by the application of 3 L/min of volatile isoflurane with pure oxygen in the induction chamber of a COMPAC5 (VetEquip) small animal anesthesia unit. After achieving a sufficient level of narcosis, the mouse was placed in a supine position on a temperature-controlled self-regulated heating system calibrated to 38 °C and fixed with stripes at all extremities. Anesthesia was reduced to a maintenance dose of 1,5 L/min isoflurane and breathing characteristics and sufficient analgesia closely monitored. The abdomen was cut open layer-by-layer cranially to create a 2-centimeter opening. Blunt retractors (FST) were placed for convenient access. Next, the cecum was grabbed with anatomical forceps to mobilize the gut to the left side of the mouse, where it was placed on a PBS-soaked sterile gauze. A second piece of gauze was used to sandwich the gut, deliberately lifting the duodenum to visualize the aorta abdominalis. A Q-tip was used to gently push the liver cranially to fully access the right renal pedicle. Under view with a surgical microscope (Carl Zeiss), sharp forceps were used to pinch retroperitoneal holes directly cranially and caudally of the pedicle. Via this access, a 100 g pressure micro serrefine (FST 18055-03) was placed on the pedicle to induce ischemia and a timer was started. The Q-tip was removed, and the packed gut swapped to the right side of the mouse to visualize the lift renal pedicle. If required, the Q-tip was used to gently push away the spleen or stomach. As before, retroperitoneal access was achieved by pinching holes with sharp forceps and another 100 g pressure micro serrefine was placed. The time between placement of both serrefines was recorded (typically < 1 min), the gut placed back in the abdominal cavity and the opening covered with the two gauze pieces. 29 min after initially starting the timer, the retractors were put in place again and the gut again mobilized and packed to visualize the right kidney. By the second after 30 min, the vascular clamp was removed, and the gut switched to the right side. After the recorded time difference, this clamp was removed as well. Reperfusion was determined visually for both sides and the gut again put back into the abdominal cavity. The peritoneum parietal and the cutis, respectively, were closed separately by continuous seams using a 6–0 Monocryl thread (Ethicon). Isoflurane application was stopped, and 1 mL of pre-warmed PBS was administered intraperitoneally. The mice were separated into pairs of two and 0.1 μg/g buprenorphine-HCl was administered every 8 h for analgesia and as required. After 48 h of rigorous observation, blood was collected by retroorbital puncture and the mice were sacrificed by cervical dislocation. The right kidney was removed to be fixed for 24 h in 4% paraformaldehyde and then transferred to 70% ethanol for storage. The left kidney was removed and snap frozen for storage.

**Total iron measurement in organs**. Measurement of the iron concentration in various organs was performed via Electrothermal Atomic Absorption Spectroscopy (ETAAS) as previously described[57]. Briefly, the tissue was digested with nitric acid at 60 °C for 12 h, after which it was adjusted to a volume of 1.5 mL with doubly distilled water. Using an atomic absorption spectrometer (Perkin-Elmer, AAnalyst 800) the iron concentration was determined against an aqueous Fe standard.

**Catalytic iron measurement in plasma**. A modified version of the bleomycin detectable iron assay originally described by Gutteridge, Rowley, and Halliwell[58] was used to measure catalytic iron levels. Briefly, the assay is based on the principle that bleomycin degrades DNA in the presence of catalytic iron in the sample, thereby producing a thiobarbituric acid (TBA) reactive substance. Upon reaction with TBA a chromogen is formed, of which the intensity was measured at 532 nm using a Beckman Coulter (DU 800) UV VIS spectrophotometer. All reagents except bleomycin were treated overnight with Chelex 100 (Bio-Rad; #1421253) resin to remove possible iron contamination. Catalytic iron levels are expressed in μmoles/l. The inter assay coefficient of variation of this assay was 389%. The lower limit of detection of catalytic iron by this assay was 0.03 μmol/L.

**Histology**. The organs were fixed in 4% paraformaldehyde, embedded in paraffin, and cut at 3 or 5 μm thickness. Subsequently, sections were stained with hematoxylin and eosin or via the Periodic acid Schiff (PAS) method. The terminal deoxynucleotidyl transferase dUTP nick end labeling (TUNEL) assay was performed according to the manufacturer's instructions (In situ cell death detection kit, TMR red—Roche). Micrographs were acquired using a Zeiss Axioscan Z.1 slide scanner (Carl Zeiss, Jenna, Germany) at 20x, 100x, 200× and 400× magnification,

with a Hamamatsu ORCA Flash4 camera (Hamamatsu Photonics) or AxioCam MRm Rev. 3 FireWire camera, via either Zen 3.1 software or AxioVision 4.5 software from Zeiss. Quantification analysis was performed using a script provided by the VIB Bioimaging Core (Ghent, Belgium) ran on QuPath-0.2.3 software.

**Serology.** The blood collected after dissection was centrifuged (10 min, 2000 g, 4 °C) to obtain plasma for further analysis. Alanine aminotransferase (ALT), aspartate aminotransferase (AST), creatine kinase (CK), lactate dehydrogenase (LDH), iron and hemolysis levels in plasma were measured in the clinical lab of Ghent University Hospital by a COBAS 8000 modular analyzer series (Roche Diagnostics, Basel, Switzerland). Medical analysis laboratory CRI Ghent analyzed creatinin, urea, ferritin (Cobas C), and troponin T (Cobas E) plasma levels. Plasma myglobin (Abcam; #ab210965), skeletal troponin C (Life Diagnostics; STNC), cardiac troponin I (Life Diagnostics; CTNI-1-US), and Myh11 (Aviva Systems Biology; #OKEH05557) levels were obtained via ELISA as per the manufacturer's instructions. Plasma cytokines levels were measured using a personalized Bio-Plex Multiplex immunoassay from Life Technologies Europe according to the manufacturer's instructions and absolute and relative blood counts were measured with Hemavet 950 from Drew Scientific.

**Colorimetric lipid peroxidation assay.** An approximation of malondialdehyde (MDA) levels was made via colorimetric measurement of 1-methyl-2-phenylindole reactive species, as previously described[59]. Of note, however, using this method no distinction could be made from the reactive species formed by 1-methyl-2-phenylindole its reaction with unrelated reactive carbonyls. Tissue analysis required prior mechanical lysing in PBS, centrifugation (10 min, 1000 g, 4 °C), and collection of the supernatant. Briefly, 100 μL of the aqueous sample was added to 325 μL of a solution of 1-methyl-2-phenylindole (Santa Cruz; #sc-253936) dissolved in a mixture of acetonitrile/methanol (3:1), with a final concentration of 10 mM 1-methyl-2-phenylindole. The reaction was then started by adding 75 μL of 37% hydrochloric acid. Upon incubation of the reaction mixture at 70 °C for 45 min, the samples were centrifuged (10 min, 15000 g, 4 °C) and the supernatant was collected. The 595 nm absorbance was measured, and the MDA concentration was determined against a standard of 1,1,3,3-tetramethoxypropane (Sigma-Aldrich; # 108383) as a source of MDA.

**Liver single-cell suspension.** Dissociation of whole livers was performed as previously described[60]. Briefly, after perfusion with PBS, livers were dissected, chopped finely, and subjected to GentleMACS dissociation followed by 20 min of incubation with 1 mg/mL Collagenase A (Roche; # 10103586001) and 10 U/mL DNase I (Roche; 10104159001) dissolved in RPMI (Gibco; #52400-025) at 37 °C, while shaking. Following a second round of GentleMACS dissociation, single-cell suspensions were filtered over a 100 μm mesh filter and centrifuged (5 min, 400 g) with an excess of PBS. Any remaining red blood cells were lysed by resuspension in ACK buffer (Lonza; #10-548E) for 3 min, after which the cells were washed in PBS, further filtered over a 40 μm mesh filter and centrifuged once more at 400 g for 5 min.

**Kidney single-cell suspension.** Following perfusion with PBS, both kidneys were dissected, chopped finely, and subjected to a 30 min enzymatic digestion with Collagenase type 1 (Sigma-Aldrich, #C-9891) dissolved in DMEM (Invitrogen; # 41965-039) at 37 °C. The remaining tissue was then gently disrupted by pipetting (25 mL pipet) and filtered over a 70 μm mesh filter. After the suspension was allowed to settle for 5 min, the upper part, not containing any fragments, was collected. The process of pipetting and filtering the suspension was repeated twice (10 mL and 5 mL pipet respectively) for the remaining settled cells in fresh DMEM. The pooled suspensions obtained were centrifuged (10 min, 1200 g) and the pellet was resuspended in ACK lysis buffer for 5 min before being centrifuged with an excess amount of DMEM once more at 1200 g for 10 min.

**Measurement of lipid ROS by flow cytometry.** Lipid ROS measurement was performed using the C11-BODIPY (581/591) probe (Molecular Probes; # D-3861)[61], which changes its fluorescent properties upon oxidation by lipid ROS molecules. Briefly, the single cell suspensions of the different organs investigated were washed in PBS and incubated in 1 mL of PBS in the presence of 0.5 μM C11-BODIPY for 15 min in 37 °C. Fluorescence intensity was measured using Fortessa LSRII in the B530 channel. DRAQ7 (0.5 μM) (Biostatus; # DR71000) was added prior to data acquisition to monitor the levels of cell death. Data analysis was performed using the FlowJo 10.7 software.

**Western blot analysis.** The different organ tissues were homogenized in NP-40 lysis buffer (10% glycerol, 1% NP-40, 200 mM NaCl, 5 mM EDTA and 10 mM Tris-HCl pH 7–7,5) supplemented with protease inhibitors leupeptin (1 mM), aprotinin (0.1 mM) and PMSF (1 mM) (Sigma-Aldrich; resp. # L2884, # A-1153, and # P-7626) and further denatured in Laemmli buffer by boiling for 10 min. Separation of proteins was performed by SDS-PAGE and the proteins were transferred to nitrocellulose membrane (Perkin Elmer; # NBA085G001EA) with semi-dry blotting. The membrane was blocked using 5% non-fat dry milk solution

in TBS buffer with 0.05% Tween20 (TBST). Incubation with primary antibody against GPX4 (1:5000 Abcam; #ab125066) and ß-Tubulin coupled to HRP (1:10000, Abcam; #ab21058) was performed O/N at 4 °C in TBST. After extensive washing, the GPX4 membrane was incubated with HRP-conjugated secondary anti-rabbit antibody (1:5000; VWR International, NA934) for 1 h in RT. Membranes were developed using Western Lighting Enhanced Chemiluminescence Substrate (Perkin Elmer; # NEL105001EA).

**Liquid chromatography–tandem mass spectrometry (LC-MS/MS) analysis.** UAMC-3203 detection in plasma, whole blood, and tissue homogenates (lysed in PBS using a Precellys 24 Tissue Homogenizer (Bertin Instruments)) was performed on an Agilent 1200 series LC system connected to a 6410 triple quadrupole mass spectrometer from Agilent Technologies (Waldbronn, Germany) with electrospray ionization (ESI) interface operated in positive ionization mode. Chromatographic separation was carried out on a Kinetex Biphenyl column (100 × 2.1 mm, 2.6 μm; Phenomenex (Utrecht, the Netherlands)). The mobile phase consisted of (A) ultrapure water with 0.1% formic acid and (B) acetonitrile/ultrapure water (90/10) with 0.1% formic acid, in gradient at 0.3 mL/min. The ESI source parameters were gas temperature 350 °C, gas flow 10 L/min, nebulizer pressure 35 psi, and capillary voltage 4000 V. Data acquisition was done in multiple reaction monitoring modes (MRM). Confirmation of UAMC-3203 was done using three MRM transitions; the most abundant transition was used as a quantifier (Q) and the other two were used as a qualifier (q). Qualifier/quantifier ratios (q/Q) were calculated for each sample and had to be within ±20% of the q/Q ratio observed in the calibrators. In addition, the retention time of the compound in samples could not deviate >10% of the retention time observed in the calibrators.

UAMC-3203 and nordiazepam-D$_5$ (Cerilliant Corporation; Texas, US) as internal standard (IS) were diluted in LC-MS grade methanol (Fisher Scientific). A volume of 100 μL sample was spiked with 20 μL IS (200 ng/mL), followed by the addition of 150 μL acetonitrile for plasma and blood. For tissue, 500 μL acetonitrile with 0.1% formic acid was added. Afterward, the mixture was vortexed (2 min, 2000 rpm) and centrifuged (10 min, 9168 g resp. 17968 gfor plasma and blood resp. tissue). The supernatant of plasma and whole blood was then transferred to a 2 mL tube with a 0.20 μm centrifugal filter (VWR, Avantor, Randor, PA, USA). The supernatant of tissue was evaporated under a stream of nitrogen at 40 °C, reconstituted in 100 μL acetonitrile/ultrapure water (90/10) with 0.1% formic acid, and transferred to a 2 mL tube with a 0.20 μm centrifugal filter. All samples were then centrifuged (5 min, 9168 g resp. 10 min, 17968 g for plasma and blood resp. tissue), after which the final extract was transferred to an autosampler vial with a glass insert. Seven-level calibration curves were prepared in blank mouse plasma or whole blood, covering a linear range from 10 ng/mL to 700 ng/mL. Five-level calibration curves were prepared in blank homogenized mouse tissue matrix covering a linear range from 20 ng/mL to 4000 ng/mL. The measured concentrations in ng/mL were further normalized using the weight of the tissue used for homogenization to obtain final concentrations of UAMC-3203 expressed in μg/g.

**Statistical analysis.** The statistical analyses of the patient data were performed with R version 3.6.1 using the ggstatsplot package version 0.5.0[62]. Survival analyses were run using the packages survival (v3.2.7) and survminer (v0.4.8). MDA and Fe$_c$ concentrations were log2 transformed with a pseudocount of 1 and the maximum MDA and Fe$_c$ concentrations per patient within the 7 study days (MDA$^{max}$) were calculated. Because of dropout and the increase of missing data throughout the study, the complete-cases data will diversify in a number of patients per day. For the MDA$^{max}$ and Fe$_c^{max}$ analysis, 176 participants were included. Associations between MDA or Fe$_c$ and SOFA, sepsis of the same day, and survival outcome at 30 days follow-up were investigated using a Spearman's rank correlation coefficient for continuous variables and a Wilcoxon–Mann–Whitney test or Kruskal–Wallis test for categorical variables. Associations between MDA and Fe$_c$ within a patient were investigated using a paired Wilcoxon signed-rank test of the Fe$_c$ levels at the time points corresponding to MDA$^{min}$ and MDA$^{max}$. The association between MDA$^{max}$ and the probability of survival was evaluated using a Log-Rank test that compared the difference between the Kaplan–Meier plots of patients with a MDA$^{max}$ < 14.25 μM resp. 16.9 versus ≥ 14.25 resp. 16.9 μM at 30 days follow-up. Finally, to assess the relative contribution of MDA$^{max}$ and Fe$_c$ to the daily hazard of death, a Cox proportional hazards model was fit with MDA$^{max}$ level, the corresponding Fe$_c$ level, age, and SOFA on the day of enrollment in the study as predictors.

Sample size estimation was done by using G power 3.1.9.7. Graphpad version 8.4.3 was used to apply the Mantel–Cox test to survival curves and to perform a two-tailed T-test or Mann–Whitney test depending on the outcome of the D'Agostino and Pearson test for normality, as indicated in the relevant panel legends. Because of the underlying Poisson distribution, a log-linear regression model of the form $g(u) = \mu_0 + \text{time}_j + \text{organ}_i + \text{organ}_i \times \text{time}_j + \varepsilon_{ijk}$, with a log link function $g(u)$ was fitted to the data in Fig. 2e–g (Genstat version 20). T statistics were used to assess the significance of the time effect (on the log scale) by pairwise comparison to the control group for each organ. Estimated mean and standard error of the mean (sem) values were formed on the scale of the response variable. When relevant, a linear fixed model of the form $y = \mu + \text{replicate}_i + \text{treatment(or genotype)}_j + \varepsilon_{ijk}$ was fitted to the data followed by pairwise T-testing to assess the

significances of pairwise comparisons between treatments or genotypes, as indicated in the individual panel legends.

Body temperatures were analyzed as repeated measurements using the method of residual maximum likelihood (REML), as implemented in Genstat version 20. Briefly, a linear mixed model (random terms underlined) of the form $y = \mu + \text{experiment}_l + \text{treatment}_i + \text{time}_j + \text{treatment}_i \times \text{time}_j + \underline{\text{mouse}_k} \times \underline{\text{time}_j}$ was fitted to the repeated measurements. The term $\text{mouse}_k \times \text{time}_j$ represents the residual error term with dependent errors because the repeated measurements are taken in the same individual, causing correlations among observations. Several covariance models were fitted to the data to account for the correlation present in the data. The autoregressive correlation model (AR) was finally selected as best covariance model to account for the correlation present in the data. The AR covariance model assumes that correlation between observations decays as the measurements are collected further apart in time. Additional options selected to get a best-fitting model included (1) times of measurement were set as equally spaced and (2) allowance of unequal variances across time. The significance of the fixed terms in the model and significance of changes in the difference between treatment effects over time were assessed using an approximate $F$-test as implemented in Genstat version 20.

**Reporting summary**. Further information on research design is available in the Nature Research Reporting Summary linked to this article.

## Data availability
The authors declare that the data supporting the findings of this study are available within the article and Supplementary Information Files. Source data are provided with this paper containing all unprocessed results and uncropped gels or blots per figure. Source data are provided with this paper.

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

## Acknowledgements

We thank the VIB Flow Core and the VIB Bioimaging Core for training, support, and access to the instrument park and are grateful for the statistical support provided by M. Vuylsteke. We thank C. Peleman for the histopathological descriptions and B. Martin (UAntwerp) for editing the article. The measurement of catalytic iron was performed at the Laboratory of Muljibhai Patel Society for Research in Nephro-Urology, Nadiad, India. We thank following institutions for funding of the project: Strategic Basic Research Foundation Flanders, IRONIX, S001522N; Research Foundation Flanders 1181917 N, 1181919 N, G0B7118N, G0C0119N, G049720N (S.V.C., E.H., T.V.B.); Excellence of Science MODEL-IDI (T.V.B., P.V.) and EOS INFLADIS (T.V.B, P.V.); Consortium of excellence at University of Antwerp INFLA-MED (M.B., K.A., T.V.B.); Industrial research Fund from University of Antwerp (M.B., K.A., T.V.B.); Industrial Research Fund from Ghent University F2012/IOF-Advanced/001(E.M., E.H.); UGent Special research fund BOF14/GOA/019 (B.W.); Foundation against cancer FAF-F/2016/865 (P.V.), FAF-C/2018/1250 (B.W., B.H., T.V.B.), F/2020/1505 (P.V.); Charcot Foundation (T.V.B.); VLIRUOS TEAM2018-01-137 (T.V.B., P.V.); Research Foundation Flanders G0C7618N, G0B718N, G.0B9620N, G0A9322N (P.V.); iBOF20/IBF/039 ATLANTIS (P.V.); FWO-SBO S001522N (T.V.B., K.A.); Flemish Institute of Biotechnology VIB (Y.S., P.V.); Methusalem BOF16/MET_V/007 (P.V.); German Research Foundation 324141047, SFB-TRR 127, SFB-TRR 205, IRTG 2251 (A.L.); Deutsche Forschungsgemeinschaft CO 291/7-1(M.C.); German Federal Ministry of Education and Research VIP + program NEUROPROTEKT 03VP04260 (M.C.); Ministry of Science and Higher Education of the Russian Federation 075-15-2019-1933 (M.C.); European Research Council under the European Union's Horizon 2020 research and innovation program GA 884754 (M.C.).

## Author contributions

S.V.C. and T.V.B. wrote the manuscript. S.V.C., B.H., E.H., K.A., and T.V.B. designed the study and experiments. S.V.C., E.V.S., I.G., B.W., B.M., W.T., S.M.C., R.R., C.D., L.L., G.D., W.W., and J.H. performed the experiments and analyzed the outcome. I.I., M.C., S.L., M.R., and A.V. provided access to and help with specific methodologies. E.M., Y.S., A.L.N.v.N., M.R., P.V., E.H., M.B., K.A., and T.V.B. provided funding for the project and M.B. was responsible for project administration. P.V., E.H., K.A., and T.V.B. gave scientific input and supervised the project. S.V.C., B.W., B.H., E.M., R.S., A.V., A.L.N.v.N., M.C., A.L., M.B., M.R., P.V., E.H., and T.V.B. revised the manuscript.

## Competing interests

T.V.B., P.V., and K.A. hold patents US9862678, WO2016075330, EP3218357, and WO2019154795 related to ferrostatin-1 analogs. M.R. and S.L. report holding United States patents (US 7,927,880 B2 19 April 2011 and US 8,192,997 B2 5 June 2012) and European patents (EP2250500B, 24 April 2013) for the methods and kit for the measurement of serum catalytic iron for early detection of acute coronary syndrome and prediction of adverse cardiac events. A.L. issued a patent for Nec-1f, an inhibitor of ferroptosis (20160943.5). M.C. is co-founder and shareholder of ROSCUE Therapeutics GmbH. The remaining authors declare no competing interests.
