## [Peer Review File · Nature Communications]

Reviewers' Comments:

Reviewer #1:

Remarks to the Author:

In this interesting manuscript, Coillie et. al. describe a possible functional link between programmed cell death by ferroptosis and the pathogenesis of multiple organ dysfunction syndrome (MODS), eventually leading to the onset of mortality in critically ill patients. The authors report a weak but significant correlation between the accumulation in plasma of catalytic iron and more so of malondialdehyde, a byproduct of lipid peroxidation, and compromised outcome of critically ill patients.

An analogue of ferrostatin 1, UAMC-3203 was developed to correct lipid peroxidation in a variety of experimental models in mice, testing to what extent ferroptosis is responsible for the development of MODS. The data obtained suggests a possible application of this new drug in the treatment of MODS induced by iron overload as well as in genetically modified mice where ferroptosis can lead to organ dysfunction and death but not in experimental models of sepsis. The authors propose that ferroptosis is a pivotal pathologic process underlying the pathogenesis of MODS, and that the novel ferrostatin analogue-UAMC-3203 developed, maybe considered as a therapeutic options in the treatment of non-septic critically ill patients. Overall the experimental design is innovative and robust, the manuscript is very well written and the data is strong but short of providing a level of irrefutable evidence to support the conclusions reached.

Perhaps the major concern relates to the pathophysiologic relevance of the experimental models of MODS used, where ferroptosis inhibition is shown to provide a clear benefit. These experimental models consist mainly on the use of genetically modified mice, where inducible gene (Gpx4) deletion elicits ferroptosis and lethality. Alternatively the authors use an experimental model whereby iron overload is imposed upon parenteral iron administration, mimicking perhaps iron poisoning. Other experimental models could be considered, such as intravascular haemolytic conditions driven for example, by red blood cell mutations.

There are smaller several points that could be addressed to strengthen the manuscript.

1. Figure 2b) An explanation for the induction of Troponin (Trop I) might be needed, given that the low level of cardiac MDA and iron accumulation.
2. Figure 2d) The pattern of the TUNEL positive cells in the liver is difficult to interpret, i.e. at the edge of the tissue.
3. Figure 3) A more detailed analyses, including MDA plasma levels, H&E and TUNEL staining of the kidney and liver could be provided in the UAMC-3203 treated Gpx4RTEKO and Gpx4HEPKO mice, respectively. This should strengthen the conclusion that UAMC-3203 inhibits lipid peroxidation in these experimental models.
4. Figure 3i,j,i,m) The nomenclature of the mice is confusing.
5. Line 83-85) The acute phase protein hepcidin should induce iron sequestration in parenchymal tissue and macrophages, which, in theory, should promote ferroptosis.
6. Line 142-144) This sentence is confusing.
7. Line 170-172. It is surprising not to see the bioanalytical profiles of UAMC-3203 in the liver, where it should exert most of its function.

Reviewer #2:

Remarks to the Author:

In this manuscript, the authors present interesting data to support their claim that targeting ferroptosis, an iron-dependent form of cell death characterized by strong lipid peroxidation, can protect against multiorgan dysfunction (MODS) and subsequent lethality. Their analysis of the plasma of a cohort of critically ill patients indeed revealed a correlation of MODS/death with higher levels of plasma iron and malondialdehyde (MDA), a product of lipid peroxidation, implying the disease relevance of ferroptosis. Further, they showed that a novel ferroptosis inhibitor they developed (UAMC-3203, an analog of known ferroptosis inhibitor, ferrostatin-1) can reduce the MODS-like symptoms in mice induced by iron-overloading, but not sepsis-like symptoms induced

by TNF or cecal ligation and puncture (CLP), which is consistent with their genetic experiments using mice with induced overexpression or knockout of GPX4, a physiological suppressor of ferroptosis. Overall this manuscript is interesting and its conclusion is consistent with the literature. However, there are also some notable weaknesses as described below:

1. Using the iron-overloading mouse model to represent general MODS is a huge overstatement. One might even question the clinical relevance of this model (injecting 300 mg/Kg FeSO₄ into mice, that is a whole body concentration of ~ 2 mM! What is the equivalent amount for an average person?). Further, for this specific model, the excessive iron-overloading can trigger ferroptosis appears to be a given, but one cannot extend the logic to assume ferroptosis is involved in MODS in general. If the authors intend to make the major claim about the MODS-ferroptosis link, they have to test this in one or two additional, more clinically relevant MODS models.

2. Although the UAMC-3203 data are convincing, its comparison in mouse models with ferrostatin-1 is not particularly informative, because it is well known that the pharmacokinetic profile of ferrostatin-1 is not good for in vivo experiments. A comparison of UAMC-3203 with liproxstatin-1 (a ferrostatin analog good for in vivo study) and vitamin E will be more meaningful.

Dear Editor,
Dear Reviewers,

We would like to cordially thank the reviewers for their positive and constructive feedback, which further helped us to improve and strengthen our findings. We have adapted the manuscript based on the suggestions of both reviewers. Adapted or inserted text has been highlighted in yellow in the revised manuscript. Please find below the answers to the issues raised by referees (marked in bolt). In summary, we have performed all requested experiments. This resulted in a lot of new data that is now included in the revised manuscript:

- Fig. 5d, g, h, i;
 - Fig. 6a, b, f;
 - Suppl. Fig. 6b, c, d; Extra PK study in rats
 - Suppl. Fig. 7; Extra phenylhydrazine (PHZ) model of intravascular hemolysis
-

Reviewer #1 (Remarks to the Author):

In this interesting manuscript, Coillie et. al. describe a possible functional link between programmed cell death by ferroptosis and the pathogenesis of multiple organ dysfunction syndrome (MODS), eventually leading to the onset of mortality in critically ill patients. The authors report a weak but significant correlation between the accumulation in plasma of catalytic iron and more so of malondialdehyde, a byproduct of lipid peroxidation, and compromised outcome of critically ill patients. An analogue of ferrostatin 1, UAMC-3203 was developed to correct lipid peroxidation in a variety of experimental models in mice, testing to what extent ferroptosis is responsible for the development of MODS. The data obtained suggests a possible application of this new drug in the treatment of MODS induced by iron overload as well as in genetically modified mice where ferroptosis can lead to organ dysfunction and death but not in experimental models of sepsis. The authors propose that ferroptosis is a pivotal pathologic process underling the pathogenesis of MODS, and that the novel ferrostatin analogue-UAMC-3203 developed, maybe considered as a therapeutic options in the treatment of non-septic critically ill patients. Overall the experimental design is innovative and robust, the manuscript is very well written and the data is strong but short of providing a level of irrefutable evidence to support the conclusions reached.

Perhaps the major concern relates to the pathophysiologic relevance of the experimental models of MODS used, where ferroptosis inhibition is shown to provide a clear benefit. These experimental models consist mainly on the use of genetically modified mice, where inducible gene (Gpx4) deletion elicits ferroptosis and lethality. Alternatively the authors use an experimental model whereby iron overload is imposed upon parenteral iron administration, mimicking perhaps iron poisoning. Other experimental models could be considered, such as intravascular haemolytic conditions driven for example, by red blood cell mutations.

➔ We share the concern of the relevance of any experimental model that should mimic MODS. Obviously, we had many discussions with the involved intensivists on which experimental models to set up to validate our candidate lead ferroptosis inhibitor. Essentially, considering the heterogeneity within e.g. septic or MODS ICU patients, no relevant experimental model exist that is representative for a broader group of patients (only emphasizing a certain aspect).

➔ As requested, we set up a model of hemolysis induced MODS, using phenylhydrazine. Phenylhydrazine induces anemia as a consequence of peroxidation of red blood cell membrane lipids (doi: 10.1016/0006-291X(78)90332-7). After optimizing the dose, we found that UAMC-3203 provides modest protection against phenylhydrazine induced MODS as indicated by the reduction in plasma LDH levels (see supplementary Fig. 7). We speculate that the reason for the modest effect is that the interaction of phenylhydrazine with red blood cells produces hydrogen peroxide. And it is generally known that lipophilic radical traps such as Fer1 or Lip1 do not protect against an excess of H₂O₂ (mM range). These inhibitors only protect against

ferroptosis in hypersensitive cells (expressing a catalytically inactive variant of GPX4), which die with a 1000-fold lower dose of H₂O₂ (doi: 10.1016/j.cell.2017.11.048).

→ We regret not being able to provide stronger data on the PHZ model, but we hope you can agree that this revised manuscript improved a lot due to the multitude of novel strong data further strengthening the main messages of the paper. Allow us to clarify, why we believe the current data support the two main conclusions of the paper:

1. Ferroptosis is a detrimental factor in MODS:

This conclusion is NOT based on the iron overdose model but is based on the patient data, which indicates an association between plasma Fe_c, excessive lipid peroxidation, the development of MODS, and an increased mortality risk.

2. UAMC-3203 is a superior ferroptosis inhibitor protecting against several models of acute single and multiple organ injury:

We show for the first time that non-canonical ferroptosis can drive tissue injury in different organs, which makes it very suitable for validating candidate lead ferroptosis inhibitors. Acute iron intoxication in patients is also known to induce MODS, often followed by death. This is a relevant and quick model to induce MODS. Despite this being a harsh model, still our Fer-analogue is lifesaving, illustrating its potency. This indicates that excessive lipid peroxidation (LPO) can contribute to MODS and that inhibitors of LPO such as UAMC-3203 could become novel treatment options for MODS patients with elevated labile iron/MDA levels.

There are smaller several points that could be addressed to strengthen the manuscript.
1. Figure 2b) An explanation for the induction of Troponin (Trop I) might be needed, given that the low level of cardiac MDA and iron accumulation.

→ We believe the reviewer refers to Trop T (not Trop I) in Figure 2b. Although Trop T is generally used as a marker of cardiac injury, Trop T can also be released from skeletal muscle upon damage. Importantly, the standard clinical test used for Trop T cannot distinguish trop T derived from cardiac or skeletal muscle. The early rise in Trop T might thus reflect early skeletal muscle injury rather than cardiac injury, in line with our findings described in extended figure 4a showing an increase in the release of skeletal troponin C rather than cardiac trop I. This has been clarified in the text.

2. Figure 2d) The pattern of the TUNEL positive cells in the liver is difficult to interpret, i.e. at the edge of the tissue.

→ The image was replaced by a more representable one which also shows cell death spread over the tissue.

3. Figure 3) A more detailed analyses, including MDA plasma levels, H&E and TUNEL staining of the kidney and liver could be provided in the UAMC-3023 treated Gpx4RTEKO and Gpx4HEPKO mice, respectively. This should strengthen the conclusion that UAMC-3203 inhibits lipid peroxidation in these experimental models.

→ The MDA plasma levels of Gpx4HEPKO mice treated with veh/fer1/3203/lip1 were provided as measured upon sacrifice when a human endpoint was reached or at the end of the experiment (Fig. 5g). In addition, H&E and TUNEL staining of these mice show the marked improvement of UAMC-3203 (or Lip1) treated mice over mice receiving vehicle or Fer1 (Fig5 i).

4. Figure 3i,j,i,m) The nomenclature of the mice is confusing.

→ Next to the official nomenclature indicating the genetic modifications of the mice, we refer to the simplified abbreviations Gpx4RTEKO and Gpx4HEPKO to remind the reader of which organ is affected in which mouse line.

5. Line 83-85) The acute phase protein hepcidin should induce iron sequestration in parenchymal tissue and macrophages, which, in theory, should promote ferroptosis.

→ Both reasonings are possible: On the one hand, removing excess iron in the circulation reduces the risk of damage due to the accumulation of extracellular catalytic reactive iron, while on the other hand this uptake of iron might make the parenchymal tissue more susceptible for ferroptosis. However, since this is a regulated response we favour the hypothesis in which by removing the iron in circulation (probably accompanied by upregulation of intracellular protection mechanism) a ferroptosis trigger (Fe_c) is taken away. Because this is all speculation, we removed this sentence in the revised manuscript.

6. Line 142-144) This sentence is confusing.

→ We have reformulated this phrase:

*“Upon acute iron overload, mice deficient in RIPK3, CYPD and PARP1 (*Ripk3*^{-/-};*Ppif*^{-/-};*Parp1*^{-/-}) only showed a mild drop in some plasma injury biomarkers compared to wild type (WT) mice (Fig. 3b,d). However, the reduction in organ damage was stronger when this mouse line was combined with overexpression of glutathione peroxidase 4 (GPX4) (*GPX4Tg*^{+/+}; Fig. 3b,d), which inhibits ferroptosis by reducing phospholipid-hydroperoxides to their alcohol form. This protective effect of GPX4 was confirmed in mice solely overexpressing GPX4 as well (*GPX4Tg*^{+/+}; Fig. 3c,e).”*

7. Line 170-172. It is surprising not to see the bioanalytical profiles of UAMC-3203 in the liver, where it should exert most of its function.

→ We have also performed a more extended PK analysis in rats and included this in the revised manuscript (Supplementary Fig. 5). Based on these results, we conclude that UAMC-3203 shows an extensive tissue distribution with good tissue-to-plasma ratio. This has been adapted in the revised manuscript.

Reviewer 2:

1. Using the iron-overloading mouse model to represent general MODS is a huge overstatement. One might even question the clinical relevance of this model (injecting 300 mg/kg $FeSO_4$ into mice, that is a whole body concentration of ~ 2 mM! What is the equivalent amount for an average person?). Further, for this specific model, the excessive iron-overloading can trigger ferroptosis appears to be a given, but one cannot extend the logic to assume ferroptosis is involved in MODS in general. If the authors intend to make the major claim about the MODS-ferroptosis link, they have to test this in one or two additional, more clinically relevant MODS models.

→ Allow us to clarify that 300mg/kg is in line with the dose reported in case reports of iron intoxication induced MODS (e.g. DOI: 10.4103/0972-5229.120326). Thus, our experimental model reflects at least this subset of patients.

→ Allow us to clarify that the claim about the MODS-ferroptosis link is not based on the experimental models but rather on the patient's data.

→ For more details, see also first reply to reviewer 1.

2. Although the UAMC-3203 data are convincing, its comparison in mouse models with ferrostatin-1 is not particularly informative, because it is well known that the pharmacokinetic profile of ferrostatin-1 is not good for in vivo experiments. A comparison of UAMC-3203 with liproxstatin-1 (a ferrostatin analog good for in vivo study) and vitamin E will be more meaningful.

→ This is a very good suggestion, but an enormous workload. Nevertheless, we have repeated all experiments including also Lip-1. The data obtained resulted in 7 novel figure panels: Fig. 5d, g, h, i and Fig. 6a,b,f. In summary, we confirmed the good efficacy of Lip1 *in vivo*, but UAMC-3203 proved to be at least equally effective, and is the only ferroptosis inhibitor that can be

dissolved in a physiologic solution (0.9% NaCl). This high solubility is crucial to allow easy application in e.g. normal saline infusions, and it even seems to improve its potency further.

Reviewers' Comments:

Reviewer #1:

Remarks to the Author:

The major concern raised in the initial round of review was and remains “..the pathophysiologic relevance of the experimental models of MODS used...”. In a clear effort to address this point the authors provide new data obtained in an experimental model of intravascular hemolysis. The data obtained shows a significant but very minimal protective effect of UAMC-3203. Unfortunately this data and its interpretation do not allow to overcome the initial major concern raised.

Reviewer #2:

Remarks to the Author:

The authors have satisfactorily addressed comments from this reviewer. The only suggestion for the final manuscript is to include a more detailed discussion on the limitation of the MODS mouse models used in the study.